# SustainGym: Reinforcement Learning Environments for Sustainable Energy Systems

**Christopher Yeh[1], Victor Li[1], Rajeev Datta[1], Julio Arroyo[1], Nicolas Christianson[1],**
**Chi Zhang[2], Yize Chen[3], Mehdi Hosseini[4], Azarang Golmohammadi[4],**
**Yuanyuan Shi[2], Yisong Yue[1], Adam Wierman[1]**
[1]California Institute of Technology
[2]University of California, San Diego
[3]Hong Kong University of Science and Technology
[4]Beyond Limits
[1]{cyeh,vhli,rdatta,jarroyoi,yyue,adamw}@caltech.edu
[2]chz056@ucsd.edu, yyshi@eng.ucsd.edu
[3]yizechen@ust.hk
[4]{mhosseini, agolmohammadi}@beyond.ai

## Abstract

The lack of standardized benchmarks for reinforcement learning (RL) in sustainability applications has made it difficult to both track progress on specific domains and identify bottlenecks for researchers to focus their efforts. In this paper, we present SustainGym, a suite of five environments designed to test the performance of RL algorithms on realistic sustainable energy system tasks, ranging from electric vehicle charging to carbon-aware data center job scheduling. The environments test RL algorithms under realistic distribution shifts as well as in multi-agent settings. We show that standard off-the-shelf RL algorithms leave significant room for improving performance and highlight the challenges ahead for introducing RL to real-world sustainability tasks.

## 1 Introduction

While reinforcement learning (RL) algorithms have demonstrated tremendous success in applications ranging from game-playing, *e.g.*, Atari and Go, to robotic control, *e.g.*, [1–3], most RL algorithms continue to only be benchmarked using toy environments—*e.g.*, OpenAI Gym [4]. These toy environments generally do not have realistic physical constraints, nor realistic environmental shifts over time. Furthermore, these environments are generally limited to single-agent systems, whereas real-world systems tend to involve coordination and/or competition between actors. The realism gap limits the reliable deployment of off-the-shelf RL algorithms in real-world systems.

Developing better RL algorithms to address these challenges requires a means of empirically benchmarking and comparing the performance of different algorithms in real-world settings. Our inspiration comes from progress in supervised machine learning (ML), where widespread adoption of breakthrough techniques was fueled by large datasets with standardized benchmarks, such as ImageNet for computer vision [5] and the GLUE benchmark for natural language processing [6]. More recently, many supervised learning datasets have been created to address specific real-world sustainability challenges, such as monitoring global progress towards sustainable development goals [7].

In this work, we introduce SustainGym, a suite of 5 RL environments that realistically model sustainability settings, summarized in Table 1:

37th Conference on Neural Information Processing Systems (NeurIPS 2023) Track on Datasets and Benchmarks.

Table 1: Summary of environments included in SustainGym and their features. The "Single agent" and "Multi-agent" rows indicate what an individual RL agent controls in that environment.

| Env | EVChargingEnv | ElectricityMarketEnv | DatacenterEnv | CogenEnv | BuildingEnv |
|---|---|---|---|---|---|
| Control task | charging rates for EV charging stations | market bids for a grid-connected battery storage system | virtual capacity curve for a carbon-aware data center | dispatch set points for turbines | heating supply for buildings |
| Modeled after | charging networks at Caltech & JPL | generic test case (IEEE RTS-GMLC) | (loosely) a Google data center | specific combined cycle gas generation plant in the U.S. | generic DoE commercial reference building models |
| Single agent | all EV charging stations | single battery system | single data center | all 4 turbine units | all buildings |
| Multi-agent | one EV charging station, cooperative | N/A | N/A | one turbine unit, cooperative | one room, cooperative |
| Actions | discrete or continuous | discrete or continuous | continuous | mixed discrete & continuous | discrete or continuous |
| Rewards | cost + $CO_2$ | cost + $CO_2$ | penalty + $CO_2$ | cost + $CO_2$ | temperature difference + energy use |
| Distribution shift | MOER, EV arrivals | MOER, load | MOER | renewable wind penetration | outdoor temperature |

- EVChargingEnv models the problem of scheduling electric vehicle (EV) charging to meet user needs while minimizing $CO_2$ emissions.
- ElectricityMarketEnv models a grid-scale battery storage system bidding into the electricity market to generate profit (through price arbitrage) and reduce $CO_2$ emissions.
- DatacenterEnv models a datacenter deciding on a "virtual capacity curve" to shift flexible jobs towards times of day with lower $CO_2$ emissions.
- CogenEnv models a combined cycle cogeneration plant producing steam and electricity to meet local demand while minimizing fuel usage and ramp costs.
- BuildingEnv models the thermal control of building energy systems to reduce the total electricity consumption while satisfying the user-specified temperature requirement.

A key feature of SustainGym environments is their support for testing RL algorithms under realistic and natural exogenous distribution shifts, which generally fall under two categories:

1. *Shifts in demand.* In each environment, RL agents choose actions to satisfy some "demand" that is often affected by the behavior of unmodeled agents. For example, in EVChargingEnv, the demand is the amount of energy that needs to be delivered to EVs that have arrived at the charging network. This demand changed significantly at the start of the COVID-19 pandemic when EV drivers changed their driving behaviors (Figure 2).
2. *Shifts in environmental parameters.* Real-world environments are rarely static, and SustainGym environments reflect changing environment parameters due to temporal and/or climate changes. For example, a battery storage controller for ElectricityMarketEnv makes decisions to minimize marginal $CO_2$ emissions, but the distribution of $CO_2$ emissions varies over time as power plants are added to or removed from the electric grid (Figure 1).

Notably, the distribution shifts reflected in SustainGym are unlike the "sim-to-real" or offline-vs-online RL distribution shifts that have been more commonly studied in the literature. The sim-to-real distribution shift comes from imperfect modeling of the environment, whereas the offline-to-online RL distribution shift is caused by a change in the policy used to generate trajectories. In contrast, the exogenous distribution shifts in SustainGym are not due to imperfect environments nor policy mismatches, but rather more fundamental changes in the transition dynamics of the Markov decision processes. Note that only the transition dynamics experience distribution shift; the state space, action space, and reward functions do not change.

Two other similar lines of work to the distribution shifts in SustainGym are nonstationary RL environments [8] and distributionally robust RL [9]. However, whereas nonstationary RL typically

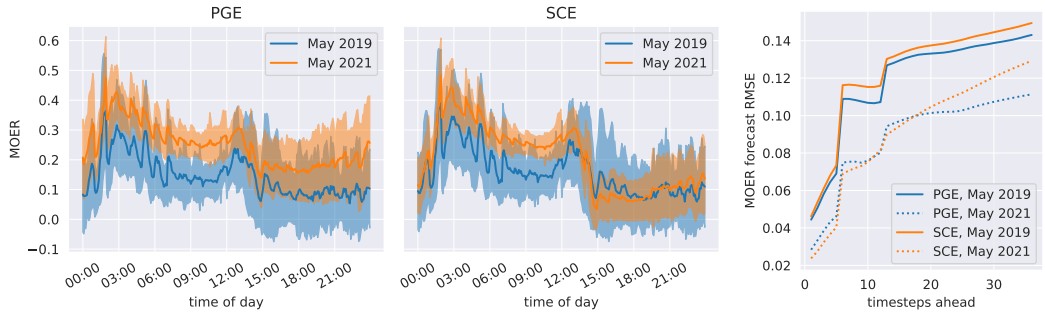

Figure 1: (left) MOER values from two different regions (a.k.a. "balancing authorities") in California, Pacific Gas & Electric (PGE) and Southern California Edison (SCE). Solid line is the mean MOER over all days in a month at a given time of day. Shaded region is ±1 std. dev. (right) MOER forecast error increases with the forecast horizon.

evaluates an RL agent's performance over the course of a changing environment, SustainGym benchmarks RL agents' ability to generalize to new (unseen) distribution shifts. Distributionally robust RL generally assumes that the set of environmental distributions are known at training time, which is not necessarily the case for the settings considered in SustainGym.

In addition to modeling realistic distribution shifts, SustainGym is distinctive for its inclusion multi-agent interactions, physical constraints, and mixtures of discrete and continuous actions, as summarized in Table 1.

To demonstrate the use of the SustainGym, we perform experiments with off-the-shelf RL algorithms. We find that these algorithms have mixed performance on SustainGym. Furthermore, we show that distribution shifts may reduce the performance of these algorithms significantly, demonstrating a need for more robust algorithms. Finally, comparisons against non-RL baselines and oracles show that RL has significant room for improvement.

Due to page constraints, the main text of this paper summarizes key design choices and experimental observations for SustainGym. Details can be found in Appendix B. Code, licenses, and instructions for using SustainGym can be found on GitHub.[1]

**Related Work.** Prior work related to SustainGym includes ConservationGym, which focuses on ecological applications [10], PowerGridWorld for power system modeling and simulation [11], and CityLearn for simulation of demand response and urban energy management [12], among others. RL environments and algorithms for both EV charging [13–15] and electricity markets [16–18] have also been released. Compared to these works, the unique aspects of SustainGym are its focus on tracking estimated $CO_2$ emissions and its ability to test RL algorithms in settings with challenging distribution shifts, physical constraints, and interactions between multiple agents. We expect SustainGym to serve as a benchmark for the progress of RL algorithm development for sustainable energy systems.

## 2 Environments

This section introduces the 5 environments in SustainGym and summarizes their design choices.

**Marginal $CO_2$ emissions.** Three environments (EVChargingEnv, ElectricityMarketEnv, DatacenterEnv) impose a cost $P_{CO_2}$ (in \$/kgCO_2) on the simulated $CO_2$ emissions induced by the actions of an agent as a result of changes in electricity consumption. To do so, our environments use data on California's historical marginal operating emissions rate (MOER, in kgCO_2/kWh), which is the increase in $CO_2$ emissions per increase in energy demand. The MOER at time $t$ is denoted $m_t \in \mathbb{R}_+$, and the forecasts generated at time $t$ for the next $k$ time steps are denoted $\hat{m}_{t:t+k-1|t} \in \mathbb{R}^k$. By default, we use $k = 36$. Figure 1 shows how MOER values and their forecasts vary across time and between different regions in California.

---

[1]`https://github.com/chrisyeh96/sustaingym/`

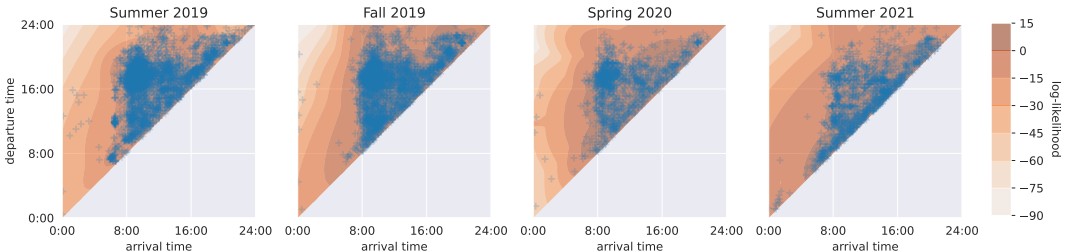

Figure 2: EV arrival vs. departure times for the Caltech EV charging network. Historical data is in blue, and log-likelihood contours from a 30-component GMM are in orange. The distribution of EV arrival and departure times changed noticeably when the COVID-19 pandemic began in early 2020.

## 2.1 EVChargingEnv

EVChargingEnv uses ACNSim [13] to simulate the charging of EVs based on actual data gathered from EV charging networks between fall 2018 and summer 2021 [19, 20]. ACNSim is a "digital twin" of actual EV charging networks at Caltech and JPL, which have $n = 54$ and 52 charging stations (abbrv. EVSEs, Electric Vehicle Supply Equipment), respectively. ACNSim accounts for nonlinear EV battery charging dynamics and unbalanced 3-phase AC power flows, and is thus very realistic. ACNSim (and therefore EVChargingEnv) can be extended to model other charging networks as well. When drivers charge their EVs, they provide an estimated time of departure and amount of energy requested. Because of network and power constraints, not all EVSEs can simultaneously provide their maximum charging rates (a.k.a. "pilot signals").

Each episode starts at midnight and runs at 5-minute time steps for 24 hours. At each time step, the agent simultaneously decides all $n$ EVSE pilot signals to be executed for the duration of that time step. Its objective is to maximize charge delivery while minimizing carbon costs and obeying the network and power constraints.

**Observation Space.** An observation at time $t$ is $s(t) = (t, d, e, m_{t-1}, \hat{m}_{t:t+k-1|t})$. $t \in [0, 1]$ is the fraction of day. $d \in \mathbb{Z}^n$ is estimated remaining duration of each EV (in # of time steps). $e \in \mathbb{R}_+^n$ is remaining energy demand of each EV (in kWh). If no EV is charging at EVSE $i$, then $d_i = 0$ and $e_i = 0$. If an EV charging at EVSE $i$ has exceeded the user-specified estimated departure time, then $d_i$ becomes negative, while $e_i$ may still be nonzero.

**Action Space.** EVChargingEnv exposes a choice of discrete actions $a(t) \in \{0, 1, 2, 3, 4\}^n$, representing pilot signals scaled down by a factor of 8, or continuous actions $a(t) \in [0, 1]^n$ representing the pilot signal normalized by the maximum signal allowed $M$ (in amps) for each EVSE. Physical infrastructure in a charging network constrains the set $\mathcal{A}_t$ of feasible actions at each time step $t$ [20]. Furthermore, the EVSEs only support discrete pilot signals, so $\mathcal{A}_t$ is nonconvex. To satisfy these physical constraints, EVChargingEnv can project an agent's action $a(t)$ into the convex hull of $\mathcal{A}_t$ and round it to the nearest allowed pilot signal, resulting in final normalized pilot signals $\tilde{a}(t)$. ACNSim processes $\tilde{a}(t)$ and returns the actual charging rate $M\bar{a} \in \mathbb{R}_+^n$ (in amps) delivered at each EVSE, as well as the remaining demand $e_i(t + 1)$.

**Reward Function.** The reward function is a sum of three components: $r(t) = p(t) - c_V(t) - c_C(t)$. The profit term $p(t)$ aims to maximize energy delivered to the EVs. The constraint violation cost $c_V(t)$ penalizes network and power constraint violations. Finally, the $CO_2$ emissions cost $c_C(t)$, which is a function of the MOER $m_t$ and charging action, aims to reduce emissions by encouraging the agent to charge EVs when the MOER is low.

**Distribution Shift.** EVChargingEnv supports real historical data as well as data sampled from a 30-component Gaussian Mixture Model (GMM) fit to historical data. We fitted GMMs to 4 disjoint historical periods, as defined in [21]. Figures 2 and 6 show the distribution of arrival and departure times in each of these 4 periods, for both the historical data as well as the GMM log-likelihoods. From these figures, it is evident that the pattern of user arrival and departure times changes over time, with the most drastic shift happening between Fall 2019 and Spring 2020, which is when the COVID-19 pandemic began.

**Multiagent Setting.** The multiagent setting features $n$ agents, each deciding the pilot signal for a single EVSE. The reward is split evenly among the agents. Each agent obtains the global observation, except that the estimated remaining durations and energy demands for other EVSEs are delayed by $t_d$ time steps.

## 2.2 `ElectricityMarketEnv`

`ElectricityMarketEnv` simulates a realtime electricity market for 33 generators and 1 80MWh battery storage system connected on a 24-bus congested transmission network based on the widely-used IEEE RTS-24 test case [22], with 5-minute settlements and load data from IEEE RTS-GMLC [23]. While `ElectricityMarketEnv` is not modeled after any particular real-world transmission network, the RTS-GMLC electricity load profile was designed to be representative of a modern transmission network located in the southwestern U.S.

All participants submit bids to the market operator (MO) at every time step. Based on the bids, the MO solves the multi-timestep security-constrained economic dispatch (SCED) problem which determines the price and amount of electricity purchased from (or sold by) each generator and battery to meet realtime electricity demand. Each episode runs for 1 day, with 5-minute time intervals. The agent controls the battery system and is rewarded for submitting bids that result in charging (buy) when prices are low, and discharging (sell) when prices and $CO_2$ emissions are high, thus performing price arbitrage.

**Observation Space.** An observation is $s(t) = (t, e, a(t - 1), x_{t-1}, p_{t-1}, l_{t-1}, \hat{l}_{t:t+k-1}, m_{t-1}, \hat{m}_{t:t+k-1|t})$. $t \in [0, 1]$ represents the time of day. $e \in \mathbb{R}_+$ is the agent's battery level (in MWh). $a(t - 1) \in \mathbb{R}_+^{2 \times k}$ is the previous action taken. $x_{t-1} \in \mathbb{R}$ is the previous dispatch (in MWh) asked of the agent. $p_{t-1} \in \mathbb{R}_+$ is the previous price experienced by the agent (in \$/MWh). $l_{t-1} \in \mathbb{R}_+$ is the previous demand experienced by the agent (in MWh), while $\hat{l}_{t:t+k-1} \in \mathbb{R}^k$ is the forecasted demand for the next $k$ steps.

**Action Space.** An agent action is a bid $a(t) = (a^c, a^d) \in \mathbb{R}_+^k \times \mathbb{R}_+^k$, representing prices (\$/MWh) that the agent is willing to pay (or receive) for charging (or discharging) per MWh of energy, for the next $k + 1$ time steps starting from time $t$. The generators are assumed to always bid their fixed true cost of generation. The environment solves the optimal dispatch problem to determine the electricity price $p_t$ and the agent's dispatch $x_t \in \mathbb{R}$, which is the amount of energy that the agent is obligated to sell into or buy from the grid within the next time step. The dispatch in turn determines the storage system's next energy level. We also provide a wrapper that discretizes the action space into 3 actions only: charge, do nothing, or discharge.

**Reward Function.** The reward function encourages the agent to maximize profit from charging decisions while minimizing associated carbon emissions. It is a sum of three components: $r(t) = r_R(t) + r_C(t) - c_T(t)$. The revenue term $r_R(t) = p_t x_t$ is the immediate revenue from the dispatch. The $CO_2$ emissions reward term $r_C(t) = P_{CO_2} m_t x_t$ represents the price of $CO_2$ emissions displaced or incurred by the battery dispatch. The terminal cost $c_T(t)$, which is nonzero only when $t = T$, encourages the battery to have the same energy level at the end of the day as when it started. We also provide an option to delay all reward signals until the terminal time (intermediate rewards are 0).

**Distribution Shift.** Distribution shift for `ElectricityMarketEnv` comes from changes in both electricity demand and MOER profiles between summer and winter months.

## 2.3 `DatacenterEnv`

`DatacenterEnv` is a simulator for carbon-aware job scheduling in datacenters, which aims to reduce the carbon emissions associated with electrcity usage in a datacenter. Carbon-aware job scheduling is premised upon two facts: (i) a significant fraction of a datacenter's workload (*e.g.*, up to 50% in some of Google datacenters [24, 25]) is comprised of low priority jobs whose execution can be delayed, and (ii) the carbon intensity of the electric grid fluctuates predictably over time. Therefore, if the execution of low priority workload is delayed to a time of day with "greener" energy, the datacenter's carbon emissions can be minimized.

`DatacenterEnv` is loosely modeled after a Google datacenter. We assume that jobs are scheduled according to a priority queue, with jobs spread evenly across the available machines. Following

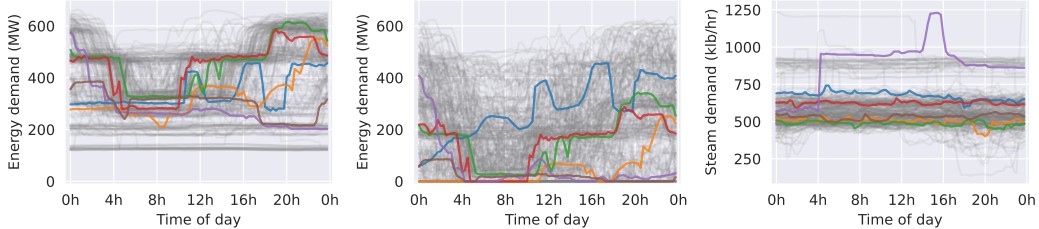

Figure 3: Electricity demand without (left) and with wind (middle), and steam demand (right) on a 15 minute basis for 253 days in `CogenEnv` dataset, with 6 random traces highlighted for each.

Radovanovic et al. [26], we implement workload execution delay by artificially limiting the total datacenter capacity with a *virtual capacity curve* (VCC) at each time step. If more jobs are enqueued than the VCC permits, then the jobs must wait until the VCC is raised high enough to allow the jobs to run. Simulation is carried out by replaying a sample of real job traces from a Google cluster from May 2019 [25]. One timestep in the environment corresponds to one hour, and each episode lasts the whole month.

**Observation Space.** An observation $s(t) = (a(t-1), d_t, n, \hat{m}_{t:t+23|t}) \in \mathbb{R}^{27}$ contains the active VCC $a(t-1)$ set from the previous time step, currently running compute load $d_t$, number of jobs waiting to be scheduled $n$, as well as the forecasted MOER for the next 24h $\hat{m}_{t:t+23|t}$.

**Action Space.** At time $t$, the agent sets the VCC, $a(t) \in [0,1]$, for the next time step. This action denotes the *fraction* of the datacenter's maximum capacity allowed to be allocated by the scheduler.

**Reward Function.** The reward consists of two components that encourage the agents to trade-off between scheduling more jobs and reducing associated carbon emissions. The first component penalizes the agent when jobs are scheduled more than 24h after they were originally submitted. The second component is a carbon emissions cost. Formally, the reward is specified as

$$r(t) = d_t \cdot m_t + \mathbf{1}_{[t\%24=0]} \max \left( 0, 0.97 w_t - C \sum_{h=0}^{23} a(t-h) \right)$$

where $d_t \cdot m_t$ is the carbon emissions, $C$ is the datacenter's maximum capacity, and $w_t$ is the total job-hours of enqueued jobs on that day.

**Distribution Shift.** The distribution shift in `DatacenterEnv` comes from changes in the MOER between 2019 and 2021.

## 2.4 `CogenEnv`

`CogenEnv` simulates the operation of a combined cycle gas power plant tasked with meeting local steam and energy demand. Conventional dispatchable generators suffer decreased efficiency as a result of frequent ramping, posing a particular challenge as increasing penetrations of variable renewables necessitate larger and more frequent ramps to ensure supply-demand balance. Thus, optimal operation of cogeneration resources requires balancing the competing objectives of minimizing fuel use, anticipating future ramp needs, and ensuring delivery of sufficient energy and steam to the grid.

While `CogenEnv` models a specific combined cycle gas generation plant in the U.S. (anonymized and location withheld for security reasons), the basic environment setup is a representative prototype of more general dispatchable resource generation control tasks, due to its complexity (the number of variables, mixed continuous/binary decisions, complementary trains of the plant all needing to be controlled together). In addition, the environment is readily modifiable to accommodate other cost structures (*e.g.*, changing the relative magnitude of the constraint penalties vs. the ramping cost).

**Observation Space.** An observation takes the form

$$s(t) = (\tau, a(t-1), T_{t:t+k}, P_{t:t+k}, H_{t:t+k}, d^p_{t:t+k}, d^q_{t:t+k}, \pi^p_{t:t+k}, \pi^f_{t:t+k}),$$

where $\tau = t/96$ is the time (normalized by number of 15 minute intervals in a day), $a(t-1)$ is the agent's previous action, $T_{t:t+k}, P_{t:t+k}$, and $H_{t:t+k}$ are current and $k$ forecast steps of temperature,

pressure, and relative humidity, respectively, $d^p_{t:t+k}$ and $d^q_{t:t+k}$ are current and $k$ forecast steps of electricity and steam demand, respectively, and $\pi^P_{t:t+k}$ and $\pi^f_{t:t+k}$ are current and $k$ forecast steps of electricity and fuel price, respectively.

**Action Space.** The action space is a vector $a(t) \in \mathbb{R}^{15}$ specifying dispatch setpoints and other auxiliary variables for all turbines in the plant. Specifically, for each of three gas turbines, the agent specifies (a) a scalar turbine electricity output, (b) a scalar heat recovery steam flow, (c) a binary evaporative cooler switch setting, and (d) a binary power augmentation switch setting. In addition, for the steam turbine, the agent specifies (a) a scalar turbine electricity output, (b) a scalar steam flow through the plant condenser, and (c) an integer number of cooling tower bays employed.

**Reward Function.** The reward function is comprised of three components:
$$r(t) = -\left(r_f(a(t); T_t, P_t, H_t) + r_r(a(t); a(t-1)) + r_c(a(t); d^p_t, d^q_t)\right).$$
$r_f(a(t); T_t, P_t, H_t)$ is the generator fuel consumption in response to dispatch $a(t)$. $r_r(a(t); a(t-1))$ is the ramp cost, captured via an $\ell_1$ norm penalty for any change in generator electricity dispatch between consecutive actions. $r_c(a(t); d^p_t, d^q_t)$ is a constraint violation penalty, penalizing any unmet electricity and steam demand, as well as any violation of the plant's dynamic operating constraints. The sum of these three components is negated to convert costs to rewards.

**Distribution Shift.** `CogenEnv` considers distribution shifts in the renewable generation profiles, and specifically, increasing penetration of wind energy. This increased variable renewable energy on the grid necessitates more frequent ramping in order to meet electricity demand, and may pose a challenge for RL algorithms trained on electricity demand traces without such variability.

**Multiagent Setting.** The multiagent setting treats each turbine unit (each of the three gas turbines and the steam turbine) as an individual agent whose action is the turbine's electricity dispatch decision and auxiliary variable settings. The negative reward of each agent is the sum of the corresponding turbine unit's fuel consumption, ramp cost, and dynamic operating constraint penalty, as well as a shared penalty for unmet electricity and steam demand that is split evenly across agents. All agents observe the global observation.

## 2.5 `BuildingEnv`

`BuildingEnv` considers the control of the heat flow in a multi-zone building so as to maintain a desired temperature setpoint. Building temperature simulation uses first-principled physics models. Users can either choose from a pre-defined list of buildings (Office small, School primary, Apartment midrise, and Office large) and three climate types and cities (San Diego, Tucson, New York) provided by the Department of Energy (DoE) Building Energy Codes Program [27] or define a customized `BuildingEnv` environment by importing any self-defined EnergyPlus building models. Each episode runs for 1 day, with 5-minute time intervals ($H = 288$, $\tau = 5/60$ hours).

**Observation Space.** For a building with $M$ indoor zones, the state $s(t) \in \mathbb{R}^{M+4}$ contains observable properties of the building environment at timestep $t$:
$$s(t) = (T_1(t), \ldots, T_M(t), T_E(t), T_G(t), Q^{\text{GHI}}(t), \bar{Q}^P(t)),$$
where $T_i(t)$ is zone $i$'s temperature at time step $t$, $\bar{Q}^P(t)$ is the heat acquisition from occupant's activities, $Q^{\text{GHI}}(t)$ is the heat gain from the solar irradiance, and $T_G(t)$ and $T_E(t)$ denote the ground and outdoor environment temperature. In practice, the agent may have access to all or part of the state variables for decision-making depending on the sensor setup. Note that the outdoor/ground temperature, room occupancy, and heat gain from solar radiance are time-varying uncontrolled variables from the environment.

**Action Space.** The action $a(t) \in [-1, 1]^M$ sets the controlled heating supplied to each of the $M$ zones, scaled to $[-1, 1]$.

**Reward Function.** The objective is to reduce energy consumption while keeping the temperature within a given comfort range. The default reward function is a weighted $\ell_2$ reward, defined as
$$r(t) = -(1 - \beta)\|a(t)\|_2 - \beta\|T^{\text{target}}(t) - T(t)\|_2$$
where $T^{\text{target}}(t) = [T^{\text{target}}_1(t), \ldots, T^{\text{target}}_M(t)]^\top$ are the target temperatures and $T(t) = [T_1(t), \ldots, T_M(t)]^\top$ are the actual zonal temperatures. `BuildingEnv` also allows users to customize reward functions by changing the weight term $\beta$ or the parameter $p$ defining the $\ell_p$ norm.

Table 2: Distribution shift experiments

| Environment | What shifts | Original setting | Shifted setting |
|---|---|---|---|
| `EVChargingEnv` | EV sessions, MOER | Summer 2019 | Summer 2021 |
| `DatacenterEnv` | MOER | May 2019 | May 2021 |
| `CogenEnv` | Wind penetration | 0 MW wind | 300 MW wind |
| `BuildingEnv` | Ambient temperature | Summer 2004 | Winter 2003 |

Users can customize the reward function to consider $CO_2$ emissions and temperature constraints such as upper and lower temperature bounds.

**Distribution Shift.** `BuildingEnv` features distribution shifts in the ambient outdoor temperature profile $T_E$ which varies with different seasons. `BuildingEnv` supports the distribution shifts due to the variation of seasons, located cities of the buildings, and can examine the challenges brought by such shifts in the RL environment.

**Multiagent Setting.** In the multiagent setting for `BuildingEnv`, each agent controls the heating action for a single zone in the building. It must coordinate with other agents to maximize overall reward. Each agent obtains the same global observation and reward.

## 3 Experiments

For each of the 5 environments in SustainGym, we implemented baseline non-RL algorithms as well as off-the-shelf RL algorithms trained using either RLLib [28] or Stable-Baselines3 (SB3) [29]. For most environments, we tested off-policy soft actor-critic (SAC) [30] and on-policy proximal policy optimization (PPO) [31]. Note that neither RLLib nor SB3 has an implementation of SAC that supports mixed discrete and continuous actions, as found in `CogenEnv`. For `EVChargingEnv`, we also tested multi-agent implementations of PPO and SAC, where the same policy is shared across agents. Non-RL algorithms tested include random policies and model predictive control (MPC), which is a model-based controller. Detailed descriptions of the implementations for each algorithm, including hyper-parameter tuning, are given in Appendix B. Finally, to test distribution shift, we trained RL agents in both "original" and "shifted" environments, then compared their performance on the shifted environment, as described in Table 2.

## 4 Discussion and Conclusion

Our experiments, shown in Figure 4, demonstrate a wide range of outcomes for off-the-shelf RL algorithms, with no single algorithm outperforming all the rest. In `EVChargingEnv`, for example, most of the RL algorithms perform no better than random actions, with the exception of multi-agent PPO with discrete actions. On `DatacenterEnv` and `BuildingEnv`, we notice a wider spread of returns across the different RL algorithms. In contrast, model-based MPC algorithms, where available, tend to perform more consistently than most RL algorithms.

In terms of distribution shift, we see a wide range of outcomes between agents trained on the original environments versus the shifted environments. Surprisingly, in `CogenEnv`, both single-agent and multi-agent policies trained on the shifted environment perform worse on the shifted environment than agents trained on the original environment. We believe this result may be due to the increased variability of shifted environment, making the shifted environment harder to learn in. In `DatacenterEnv`, the shift in MOER values shows essentially no effect on agent performance. In `EVChargingEnv`, agents trained on the shifted environment generally perform slightly better than agents trained on the original environment. In `BuildingEnv`, agents trained on the shifted environment perform much better.

These results highlight that the distribution shifts present in SustainGym environments provide substantial opportunities for future research, including robust RL algorithms [32] as well as online learning under distribution shift. Developing RL algorithms that are robust to these natural distribution shifts will be critical for deploying RL in the real-world high-impact sustainability settings such as those modeled by SustainGym environments.

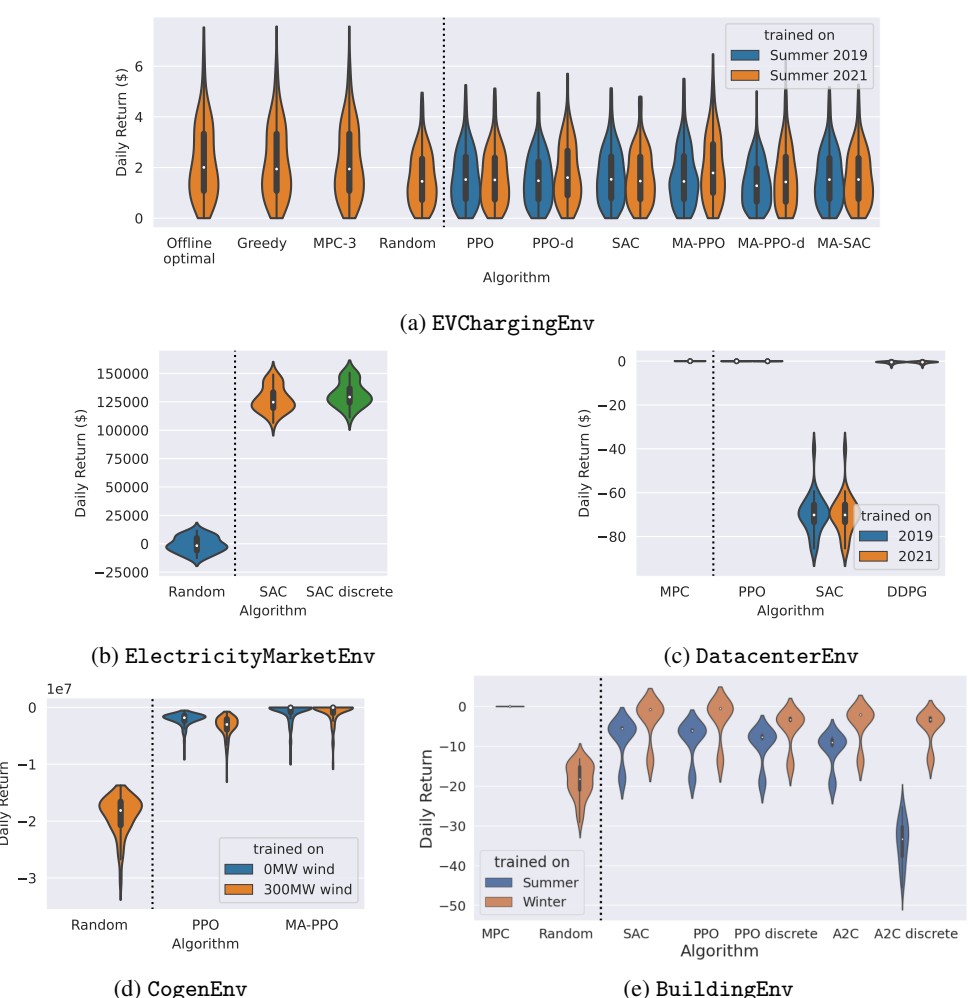

Figure 4: Experimental results for all 5 environments comparing performance on the shifted environment between RL algorithms trained on the original environment (blue) and RL algorithms trained on the shifted environment (orange). Policies using discretized actions are indicated with "-d", and multi-agent policies are prefixed with "MA-". For a complete description of the experiments, please see Appendix B.

**Multi-agent RL.** SustainGym is currently designed to support multi-agent RL in 3 environments, with the goal of upgrading all environments with multi-agent support in the future. In the two environments for which we tested multi-agent RL policies (`EVChargingEnv` and `CogenEnv`), the multi-agent PPO (MA-PPO) policies out-performed all other RL policies. We suspect that this may be because the action spaces in these environments factorize well across agents, so the multi-agent policies can learn more efficiently. Furthermore, we suspect that multi-agent policies may have the potential of performing better under distribution shift, since their environments are naturally non-stationary during training. We notice this to be true for both `EVChargingEnv` and `CogenEnv`: of the RL policies trained on the original environments, the multi-agent policies performed best when tested on the shifted environment.

**Future work.** SustainGym is under active development, with several key directions of future work:

- Comprehensive support for multi-agent RL. Currently, only 3 out of the 5 environments support multi-agent RL. We are working to extend the two other environments `ElectricityMarketEnv` and `DatacenterEnv` to the multi-agent RL setting. For `ElectricityMarketEnv`, we plan on introducing multiple batteries into the same transmission network, each controlled by separate competing agents. This will be the first

competitive multi-agent RL environment in SustainGym. (All other environments feature cooperative multi-agent RL.) For `DatacenterEnv`, we plan on introducing multiple datacenters spread across geographic regions to enable both temporal and geographic carbon-aware load shifting. Each datacenter would be its own agent.

- Different degrees of distribution shift. Currently, SustainGym environments feature a binary choice of distribution shift: an original environment, and a shifted environment. We plan on introducing more settings with varying degrees of distribution shift.
- More environments. We welcome new environment ideas and contributions to SustainGym and are working with potential collaborators to extend the scope of environments.

**Limitations**    We conclude by acknowledging general limitations of SustainGym. First, SustainGym only captures very limited dimensions of sustainability (*i.e.*, energy and $CO_2$ emissions) and does not account for other aspects such as water usage and other pollutants associated with energy production. We welcome collaboration with experts in these other sustainability domains to help us improve the sustainability mission of SustainGym. Second, SustainGym is limited in the types of distribution shifts that are considered. Finally, while SustainGym environments have been designed to be reasonably representative of various sustainable energy settings, there is inevitably a gap between SustainGym simulations and actual hardware systems. A detailed discussion of the representativeness, generalizability, and limitations of each environment can be found in Appendix B.

## Acknowledgments and Disclosure of Funding

We would like to acknowledge Steven Low, Tongxin Li, Zachary Lee, Lucien Werner, Zaiwei Chen, Ivan Jimenez, Pieter Van Santvliet, and Ameera Abdelaziz for their feedback and input during the preparation of SustainGym. The model underlying `CogenEnv` was developed in partnership with Beyond Limits and Enexsa. Funding for this project comes from a variety of sources, including NSF awards CNS-2146814, CPS-2136197, CNS-2106403, EPCN-2200692, and NGSDI-2105648; an NSF Graduate Research Fellowship; an Amazon AI4Science Fellowship; the Resnick Sustainability Institute; Hellman Fellowship; Amazon Web Services; and Beyond Limits. This work is partially supported by the Department of the Air Force under Air Force Contract No. FA8702-15-D-0001. Any opinions, findings, conclusions or recommendations expressed in this material are those of the author(s) and do not necessarily reflect the views of the Department of the Air Force. Notwithstanding any copyright notice, U.S. Government rights in this work are defined by DFARS 252.227-7013 or DFARS 252.227-7014 as detailed above. Use of this work other than as specifically authorized by the U.S. Government may violate any copyrights that exist in this work.

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
