## Appendix

## A    Metadata

### A.1    Hosting and maintenance

SustainGym is hosted as a Python package on GitHub and PyPI. SustainGym is semantically versioned, and the version accompanying this publication is designated v1.0.

A guide for contributors is available in the GitHub repo.

The following authors are the maintainers for each environment in SustainGym:

- Christopher Yeh: `EVChargingEnv`, `ElectricityMarketEnv`, `DatacenterEnv`
- Nicolas Christianson: `CogenEnv`
- Chi Zhang: `BuildingEnv`

### A.2    Licenses and responsibility

SustainGym as a whole is released under a CC BY 4.0 license.[2]  However, the `CogenEnv` and accompanying code is released under a more restrictive CC BY-NC-SA 4.0 license,[3] per the terms of the model provider Enexsa. These licenses are available in our GitHub repo.

The authors bear all responsibility in case of violation of rights. SustainGym does not contain any personally identifiable data, nor any offensive content.

### A.3    Compute requirements

The experiments for this paper were run on Amazon AWS virtual machines, Google Colab notebooks, as well as a Caltech internal compute cluster. Most of these compute resources featured NVIDIA GPUs.

### A.4    Intended uses

SustainGym is intended as a testbed for RL algorithms in sustainability-focused energy systems. While significant efforts have been undertaken to make the SustainGym environments closely match real-world systems, performance on SustainGym environments is not a guarantee of performance on real-world systems.

## B    Environment and Experiment Details

### B.1    EVChargingEnv

**Assumptions.**    In addition to the description given in Section 2.1, `EVChargingEnv` makes the following assumptions:

- EVs staying overnight can be ignored. Upon the start of each episode, we assume the garage is empty. Analysis of historical traces on the adaptive charging network show that at most 12 cars at one time stay through midnight. Because this number is small compared to the number of stations, we do not expect the assumption of an empty garage at the start of the day to significantly impact the accuracy of the environment in representing real-world settings.
- All EVs have identical batteries. Because ACN-Data does not include data on the model of each EV, yet ACN-Sim models battery charging dynamics which vary based on battery capacity, we assume that all EVs contain 100 kWh batteries and come with an initial state of charge such that if the EV were charged to the amount of the user-requested energy, the battery would be full. Energies requested over 100 kWh are capped at 100 kWh. We use the `Linear2StageBattery` battery model within ACN-Sim.

---

[2]https://creativecommons.org/licenses/by/4.0/
[3]https://creativecommons.org/licenses/by-nc-sa/4.0/

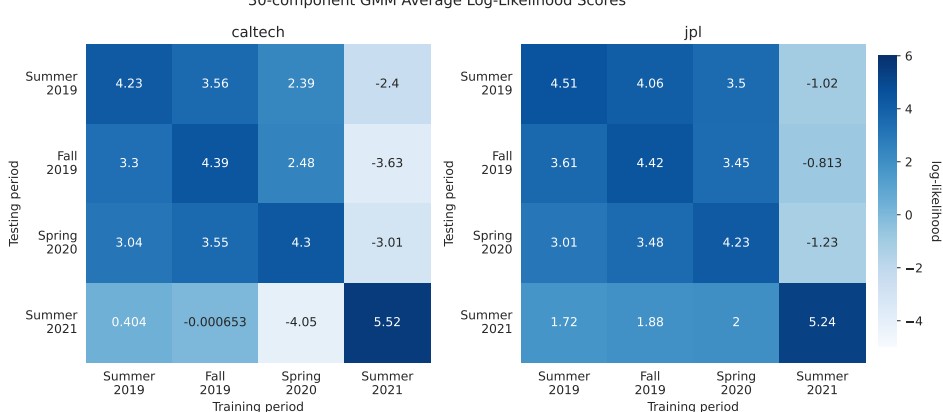

Figure 5: Log-likelihoods of data from testing periods (y-axis) for GMMs fitted on a training period (x-axis) using 30 components for the Caltech (left) and JPL (right) sites. A higher score implies a better fit. In all cases, GMMs scored highest on the period it was trained on. The significant drop in log-likelihood scores off-diagonal is indicative of distribution shift.

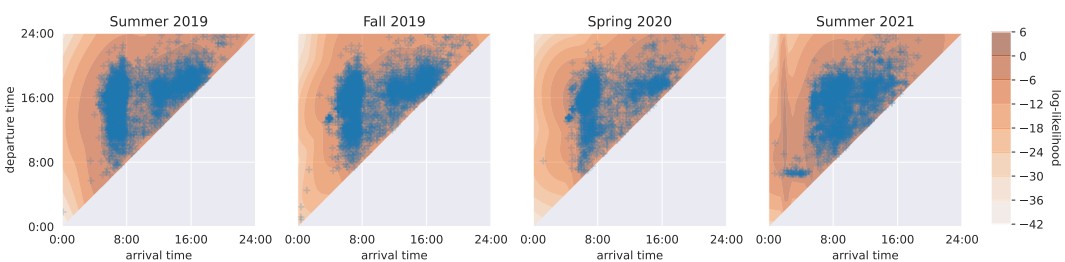

Figure 6: Like Figure 2, but for the JPL charging network.

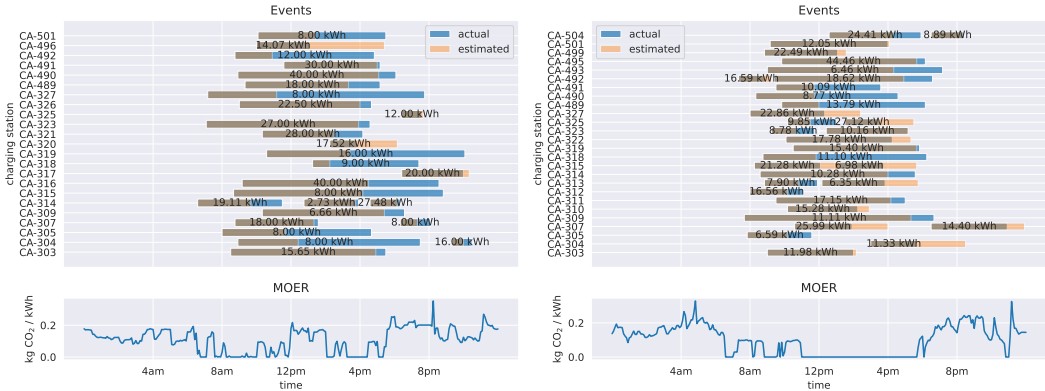

Figure 7: (left) A real full day's charging sessions from the Caltech charging network. (right) A randomly sampled trace of charging sessions from the GMMs trained on the Caltech historical data.

Thus, each historical charging session includes time of EV arrival, estimated departure, actual departure, energy delivered, and EVSE ID.

**Feasible action space.** Let $\mathcal{L}$ denote physical infrastructure resources (e.g., transformers or breakers) in the charging network. These infrastructure resources are characterized by $(A, \phi, c, v)$ where $A \in \mathbb{R}^{|\mathcal{L}| \times n}$ accounts for the charging network layout, $\phi \in \mathbb{R}^n$ is the voltage phase angle of each EVSE, $c \in \mathbb{R}^{|\mathcal{L}|}$ is the capacity limit for each resource, and $v \in \mathbb{R}_+$ is the EVSE voltage (in kV). Along with the demand at each EVSE, $(A, \phi, c)$ defines the time-dependent set of valid actions $\mathcal{A}_t$. Lastly, let $M = 32$ denote the maximum allowed pilot signal (in amps) for each EVSE.

When an agent gives the environment its desired normalized pilot signals $a(t) \in [0, 1]$, the projection into the convex hull of $\mathcal{A}_t$ is performed by solving the following convex optimization problem:

$$
\begin{aligned}
\min_{\hat{a} \in \mathbb{R}^n} \quad & \|a(t) - \hat{a}\|_2 & & \text{(1a)} \\
\text{s.t.} \quad & 0 \le \hat{a}_i \le 1 & \forall i \in \{1, \ldots, n\} & \quad \text{(1b)} \\
& M\hat{a}_i v\tau \le e_i & \forall i \in \{1, \ldots, n\} & \quad \text{(1c)} \\
& \underbrace{\left| \sum_{i=1}^{n} A_{li} M\hat{a}_i e^{j\phi_i} \right|}_{|I_l|} \le c_l & \forall l \in \mathcal{L} & \quad \text{(1d)}
\end{aligned}
$$

Here, $j = \sqrt{-1}$ is the imaginary number and $I_l$ is the aggregate current through constraint $l \in \mathcal{L}$. The quantity $M\hat{a}_i v\tau$ computes the energy (in kWh) to be charged from EVSE $i$ during the next time step. The continuous pilot signals $M\hat{a}$ are then rounded to the set of discrete pilot signals supported by each EVSE, resulting in the final pilot signals $M\tilde{a}(t)$.

**Reward function.** The profit term $p(t)$ aims to maximize the amount of energy delivered to the EVs. Let $\pi \in \mathbb{R}_+$ denote a fixed marginal profit (in \$/kWh) that the EV charging network earns for each unit of energy delivered and $v \in \mathbb{R}_+$ be the voltage (in kV) of the charging network. Then,

$$
p(t) = \pi \sum_{i=1}^{n} M\bar{a}_i v\tau = \pi M\bar{a}^\top \mathbf{1} v\tau
$$

where $\mathbf{1}$ denotes a vector of all ones. The constraint violation cost $c_V(t)$ aims to reduce physical constraint violations and encourage the agent's action $a(t)$ to be in $\mathcal{A}_t$. We penalize violation costs with a weight $\lambda_V$ (in \$/kWh), so

$$
c_V(t) = \lambda_V \sum_{l \in \mathcal{L}} \max\{I_l(t) - c_l, 0\} v\tau
$$

where $I_l$ is the electrical current and $c_l$ is the maximum current capacity at resource $l$. By default, we set $\lambda_V = 0.01$.

Finally, the $CO_2$ emissions cost $c_C(t)$ aims to reduce emissions by encouraging the agent to charge EVs when the MOER is low. We have

$$
c_C(t) = P_{CO_2} m_t \sum_{i=1}^{n} M\bar{a}_i v\tau = P_{CO_2} m_t M\bar{a}^\top \mathbf{1} v\tau
$$

**Model predictive control (MPC).** As a baseline non-RL algorithm, we consider a model predictive control (MPC) controller similar to what is proposed in [20]. Let $w \le k$ denote the length of the lookahead window (up to the number of MOER forecast steps $k$). Then, at every time step $t$, the MPC controller solves the following optimization problem:

$$\max_{a^0,\dots,a^{w-1}\in\mathbb{R}^n} \quad \sum_{k=0}^{w-1}(\pi - P_{\mathrm{CO_2}}\hat{m}_{t+k|t})Ma^{k\top}\mathbf{1}v\tau \tag{2a}$$

$$\text{s.t.} \quad 0 \preceq a^k \preceq 1 \qquad\qquad \forall k \in \{0,\dots,w-1\} \tag{2b}$$

$$\sum_{k=0}^{w-1}Ma^k v\tau \preceq e \tag{2c}$$

$$a_i^k = 0 \qquad\qquad \forall i \in \{1,\dots,n\},\ k \geq d_i \tag{2d}$$

$$\underbrace{\left|\sum_{i=1}^{n}A_{li}Ma_i^k e^{j\phi_i}\right|}_{|I_l|} \leq c_l, \qquad\qquad \forall l \in \mathcal{L},\ k \in \{0,\dots,w-1\} \tag{2e}$$

The optimization problem plans pilot signals for the next $w$ time steps to maximize revenue and minimize forecasted carbon emissions cost. (2c) ensures that the pilot signals do not over-charge the EVs. (2d) prevents charging EVs after their estimated departure times. (2e) is the usual infrastructure constraint.

Only the first planned pilot signal $Ma^0$ is used. It is rounded to the set of discrete pilot signals supported by each EVSE, resulting in the final pilot signal $M\tilde{a}(t)$. On the next time step, the MPC algorithm resolves the optimization problem. Figure 4a shows the distribution of returns for "MPC-3", which sets $w = 3$.

**Training.** The entire summer 2021 period [21] was selected as the testing period. We performed three splits: 1) by RL algorithm (PPO and multi-agent PPO vs. SAC and multi-agent SAC), 2) continuous vs. discrete action space, and 3) training data generated by GMMs based on out-of-distribution data (summer 2019) vs. in-distribution data (summer 2021). In each of the four cases, we used action projection during training and testing, eliminating costs of network violations. We tested 3 learning rates for each algorithm: PPO (5e-6, 5e-5, 5e-4) and SAC (1e-2, 1e-3, 1e-4). These learning rates were chosen as the default RLLib learning rates for each algorithm, scaled by factors of 10, 1, and 0.1. For each learning rate, three different random seeds were tested, and the model with the best training performance was selected.

**Representativeness and Generalizability** As mentioned in Section 2.1, EVChargingEnv is based on the actual EV charging networks in place at Caltech and the Jet Propulsion Laboratory (JPL), both located in Pasadena, California, U.S.A. EVChargingEnv uses a "digital twin" of these networks, called ACN-Sim [13], as well as real historical EV charging data [19], in the simulation of the RL environment. While different EV charging networks may have different constraints, the adaptive EV charging problem is similar across all networks (*i.e.*, deciding pilot signals for EVSEs to maximize energy delivery while minimizing costs and satisfying network and power constraints). In this way, EVChargingEnv is very representative of this type of control task. Furthermore, the code for ACN-Sim is open-source and well-documented,[4] which allows EVChargingEnv to be extended to model other charging networks as well.

**Limitations** As mentioned under "Assumptions" above, EVChargingEnv uses some simplifying assumptions, in part because of limited data availability. Furthermore, the reward function in EVChargingEnv currently assigns a fixed marginal profit per unit of energy delivered to an EV, whereas larger EV charging networks are often affected by time-varying wholesale electricity prices. Finally, electricity grid failures and communication failures are not modeled in EVChargingEnv.

### B.2 ElectricityMarketEnv

**System Model and Motivation.** ElectricityMarketEnv aims to simulate grid-scale battery storage systems participating in an energy market for a regional transmission network. A battery has

---

[4]https://acnportal.readthedocs.io/

Table 3: `ElectricityMarketEnv` parameters. The $k$-th entry of a vector $\mathbf{g}_t$ is denoted $g_{k,t}$.

| Param. | Domain | Unit | Description |
|---|---|---|---|
| $t$ | $[1, \ldots, T]$ | 5 min | index of real-time interval (usually 5-15 min) |
| $T$ | $\mathbb{Z}_+$ | | number of periods in each epoch, typically 1 day for 5 min intervals |
| $h$ | $\{0, 1, \ldots, T\}$ | | length of the lookahead horizon |
| $\tau$ | $[t, \ldots, t+h]$ | 5 min | index of the interval in each multi-interval lookahead subproblem |
| $N$ | $\mathbb{Z}_+$ | | number of nodes in network |
| $M$ | $\mathbb{Z}_+$ | | number of lines in the network |
| $N_G$ | $\mathbb{Z}_+$ | | number of generators |
| $N_D$ | $\mathbb{Z}_+$ | | number of loads |
| $N_B$ | $\mathbb{Z}_+$ | | number of batteries |
| $i$ | $[1, \ldots, N]$ | | index of node |
| $j$ | $[1, \ldots, M]$ | | index of line |
| $k$ | $[1, \ldots, N_G, N_D, N_B]$ | | index of generator, load, or battery |
| $B$ | $\mathbb{R}_+^{M \times M}$ | $\Omega^{-1}$ | network susceptance matrix |
| $C$ | $\{-1, 0, 1\}^{N \times M}$ | | node-line network incidence matrix |
| $\Sigma$ | $\{-1, 0, 1\}^{N \times (N_D + N_G + 2N_B)}$ | | participant-to-node mapping matrix |
| $H$ | $\mathbb{R}^{M \times N}$ | | generation shift factor matrix |
| $\mathbf{f}^{\max}$ | $\mathbb{R}_+^M$ | MVA | maximum power flow along each line |
| $\hat{\mathbf{d}}_t$ | $\mathbb{R}^{N_D}$ | MW | vector of predicted (average) load in interval $t$ |
| $\underline{\mathbf{g}}, \overline{\mathbf{g}}$ | $\mathbb{R}_+^{N_G}$ | MW | min/max generation limits (assumed to be time invariant) |
| $\overline{\mathbf{b}^{\mathrm{c}}}, \overline{\mathbf{b}^{\mathrm{d}}}$ | $\mathbb{R}_+^{N_B}$ | MW | max charge/discharge limits (assumed to be time invariant) |
| $\underline{\mathbf{x}}, \overline{\mathbf{x}}$ | $\mathbb{R}_+^{N_B}$ | MWh | min/max state of charge limits (assumed to be time invariant) |
| $\mathbf{x}^{\mathrm{f}}$ | $\mathbb{R}_+^{N_B}$ | MWh | final state of charge required at interval $T$ |
| $\mathbf{x}^0$ | $\mathbb{R}_+^{N_B}$ | MWh | initial state of charge at beginning of epoch |
| $\eta_k^{\mathrm{c}}, \eta_k^{\mathrm{d}}$ | $(0, 1]$ | | charge and discharge efficiency of battery $k$ |
| $c_{k,t}^{\mathrm{g}}$ | $\mathbb{R}_+$ | \$/MWh | marginal cost of generator $k$ in interval $t$ |
| $c_{k,t}^{\mathrm{b,c}}$ | $\mathbb{R}_+$ | \$/MWh | marginal cost of battery $k$ charging in interval $t$ |
| $c_{k,t}^{\mathrm{b,d}}$ | $\mathbb{R}_+$ | \$/MWh | marginal cost of battery $k$ discharging in interval $t$ |

two objectives: make profit from price arbitrage (buy low, sell high) while minimizing its associated $CO_2$ emissions. The $CO_2$ emissions are accounted for when both charging and discharging:

- When a battery charges energy from the grid, it incurs the $CO_2$ emissions associated with grid's current marginal emissions rate (MOER).
- When a battery discharges energy into the grid, it offsets other generators that would have had to supply more electricity, thereby reducing $CO_2$ emissions commensurate with the grid's current MOER.

Network congestion refers to physical limits on transmission line capacity that prevent a generator from supplying electricity to a load at an arbitrary bus (a.k.a. "node") on the network. Such constraints lead to inefficiencies but are present in almost all real-world transmission networks. Thus, modeling these transmission constraints is important for ensuring electricity grid reliability.

The parameters of `ElectricityMarketEnv` are listed in Table 3. The default environment is based on the Region 1 in the IEEE Reliability Test System (RTS) [23], with 5-minute settlements ($\delta = 5/60$ hours), an episode length of $T = 288$ (1 day), and a lookahead horizon of $h = 36$ steps (3 hours). The IEEE RTS network features $N = 24$ nodes (a.k.a. buses) connected with $M = 38$ lines. Connected to the network are $N_G = 33$ conventional thermal generators and $N_D = 17$ loads. The network is slightly modified to add a single ($N_B = 1$) 80MWh battery system located at bus 11, with maximum charge/discharge rate $\overline{\mathbf{b}^{\mathrm{c}}} = \overline{\mathbf{b}^{\mathrm{d}}} = 20$MW, and initial and final state of charge (SOC)

$\underline{\mathbf{x}} = 0$, $\mathbf{x}^0 = \mathbf{x}^f = 40$MWh. Charge/discharge efficiency is set to $\eta^{\mathrm{c}} = \eta^{\mathrm{d}} = 0.95$, and the minimum generation limit for all generators is changed to $\underline{g} = 0$.

We assume that an independent system operator (ISO) solves standard security-constrained economic dispatch (SCED) with the DC (decoupled) power flow equations. "Security-constrained" means that line limits are respected. The formulation has the following features:

- 3 classes of market participants: generators, loads, and storage
- DC PF equations
- Line flows and constraints
- Linear cost functions
- Nodal demand timeseries

The formulation lacks the following features that may be considered important in a practical implementation of SCED:

- Unit commitment for thermal units
- Line and generator outage contingencies, *e.g.*, $N - 1$ security
- Operating reserves
- Piecewise linear and quadratic generator cost functions
- Flexible, curtailable load
- Renewable generators
- Generator ramping constraints

**Battery storage.** There are $N_B$ batteries in the system, of which only one is seeking to learn an optimal bidding strategy. Assign index $k = 1$ to this unit.

The bid must satisfy power bounds and non-simultaneity. The strategic battery must manage its own state of charge. The other $B - 1$ batteries are assumed to be managed by the ISO. We do not consider the possibility of simultaneous charge/discharge.

The state of charge evolution of each battery is linear in charge/discharge efficiency parameters $\eta_i^{\mathrm{c}}$, $\eta_i^{\mathrm{d}}$. The charge/discharge limits and the SOC constraints are assumed to be known by the system operator for all units and are not part of the strategic agent's bid. The initial state of charge in each interval is given by the solution from the previous interval's lookahead optimization.

**Generation and load.** There are $N_G$ generators in the system, with potentially multiple generators at each node. We do not consider ramping limits on generators or non-convexities from start-up and shutdown costs and constraints. The generators are constrained between their minimum ($\underline{\mathbf{g}} = 0$) and maximum production limits $\overline{\mathbf{g}}$. It is assumed that a feasible unit commitment exists for the given $\underline{\mathbf{g}}$ and $\overline{\mathbf{g}}$.

The load vector $\mathbf{d}_t \in \mathbb{R}^{N_D}$ is taken to be fixed (inflexible load). The demand prediction for $\tau > t$ is given by $\hat{\mathbf{d}}_\tau$, provided by an hourly day-ahead forecast.

Table 4: Definitions of problem variables. Variables are optimized in the economic dispatch problem.

| Variable | Domain | Unit | Description |
|---|---|---|---|
| $\mathbf{p}_t$ | $\mathbb{R}^N$ | MW | vector of nodal power injections in interval $t$ |
| $\mathbf{d}_t$ | $\mathbb{R}^{N_D}$ | MW | vector of load power injections in interval $t$ |
| $\mathbf{g}_t$ | $\mathbb{R}^{N_G}$ | MW | vector of generator power injections in interval $t$ |
| $\mathbf{b}_t^{\mathrm{c}}$ | $\mathbb{R}_+^{N_B}$ | MW | vector of battery charging power injections in interval $t$ |
| $\mathbf{b}_t^{\mathrm{d}}$ | $\mathbb{R}_+^{N_B}$ | MW | vector of battery discharging power injections in interval $t$ |
| $\mathbf{x}_t$ | $\mathbb{R}_+^{N_B}$ | MWh | vector of battery states of charge in interval $t$ |
| $\lambda_t$ | $\mathbb{R}$ | | Lagrange multiplier corresponding to the power balance constraint $\mathbf{1}^\top \mathbf{p}_t = 0$ |
| $\boldsymbol{\mu}_t^{\pm}$ | $\mathbb{R}^M$ | | Lagrange multiplier vectors corresponding to upper/lower line limit constraints |
| $\boldsymbol{\pi}_t$ | $\mathbb{R}^N$ | \$/MWh | market-clearing nodal price vector for interval $t$ |

**Powerflow equations and line limits.** The line susceptance matrix is $B = \text{diag}(B_1, \ldots, B_M) \in \mathbb{R}_+^{M \times M}$ where $B_j \in \mathbb{R}$ is the susceptance of line $j$.

The matrix $C \in \{-1, 0, 1\}^{N \times M}$ is node-edge incidence matrix of the graph.

The slack bus of the network is assigned WLOG to node index $i = 1$. By convention the voltage angle of this node is fixed:

$$\theta_{1,t} = 0 \quad \forall t.$$

Given a vector of nodal real power injections $\mathbf{p}_t \in \mathbb{R}^N$, the power flows along lines in the network $\mathbf{f}_t \in \mathbb{R}^M$ is given by $\mathbf{f}_t = H\mathbf{p}_t$ with the generation shift matrix $H \in \mathbb{R}^{M \times N}$ defined as $H := BC^\top \left(CBC^\top\right)^\dagger$.

**Economic dispatch problem.** We implement a version of SCED called multi-interval lookahead real-time economic dispatch. In this version of the market clearing, the systems operator seeks to clear the market sequentially for each timestep (*e.g.*, every 5 mins), optimizing over the current interval $t$ plus a lookahead horizon of $h$ additional intervals. Only the solution from interval $t$ is retained; the remaining $h$ decisions are advisory. In systems with intertemporal constraints (*e.g.*, energy storage, unit commitment, ramping), it is necessary to perform this multi-interval dispatch to improve *ex-post* optimality and as well as to retain feasibility. In practice, North American ISOs all solve a version of multi-interval dispatch.

For interval $t$, the multi-interval SCED problem is

$$\min_{\mathbf{g}_t, \mathbf{b}_t^c, \mathbf{b}_t^d} \quad \delta \sum_{\tau=t}^{t+h} \left[ \sum_{k=1}^{N_g} c_{k,\tau}^g g_{k,\tau} + \sum_{k=1}^{N_b} c_{k,\tau}^{b,d} b_{k,\tau}^d - c_{k,\tau}^{b,c} b_{k,\tau}^c \right] \tag{3a}$$

$$\text{s.t.} \quad \Sigma[\mathbf{d}_\tau^\top, \mathbf{g}_\tau^\top, \mathbf{b}_\tau^{c\top}, \mathbf{b}_\tau^{d\top}]^\top = \mathbf{p}_\tau \qquad \forall \tau = t, \ldots, t+h \tag{3b}$$

$$\lambda_\tau \perp \quad \mathbf{1}^\top \mathbf{p}_\tau \delta = 0 \qquad \forall \tau = t, \ldots, t+h \tag{3c}$$

$$\boldsymbol{\mu}_\tau^+, \boldsymbol{\mu}_\tau^- \perp \quad |H\mathbf{p}_t| \leq \mathbf{f}^{\max} \qquad \forall \tau = t, \ldots, t+h \tag{3d}$$

$$\mathbf{d}_\tau = \hat{\mathbf{d}}_\tau \qquad \forall \tau = t, \ldots, t+h \tag{3e}$$

$$\underline{\mathbf{g}} \leq \mathbf{g}_\tau \leq \overline{\mathbf{g}} \qquad \forall \tau = t, \ldots, t+h \tag{3f}$$

$$\mathbf{0} \leq \mathbf{b}_\tau^c \leq \overline{\mathbf{b}^c} \qquad \forall \tau = t, \ldots, t+h \tag{3g}$$

$$\mathbf{0} \leq \mathbf{b}_\tau^d \leq \overline{\mathbf{b}^d} \qquad \forall \tau = t, \ldots, t+h \tag{3h}$$

$$\mathbf{x}_\tau = \mathbf{x}_{\tau-1} + \text{diag}(\boldsymbol{\eta}^c)\mathbf{b}_\tau^c \delta - \text{diag}(\boldsymbol{\eta}^d)^{-1}\mathbf{b}_\tau^d \delta \qquad \forall \tau = t, \ldots, t+h \tag{3i}$$

$$\underline{\mathbf{x}} \leq \mathbf{x}_\tau \leq \overline{\mathbf{x}} \qquad \forall \tau = t, \ldots, t+h \tag{3j}$$

$$\mathbf{x}_{t-1} = \mathbf{x}_{t-1}^* \tag{3k}$$

$$\mathbf{x}_{t+h} = \mathbf{x}^f \tag{3l}$$

Here, $\mathbf{x}_{t-1}^*$ is the state of charge decision from the previous interval. The constraint says that the current state of charge is whatever the battery was charged/discharged to in the previous dispatch.

In order to ensure that the battery is never simultaneously charging and discharging (*i.e.*, for every $k \in \{1, \ldots, N_b\}$ and $\tau \in \{t, \ldots, t+h\}$, at most one of $b_{k,\tau}^c$ and $b_{k,\tau}^d$ should be nonzero), we can instead formulate the problem as a mixed-integer linear program:

$$\min_{\mathbf{g}_t, \mathbf{b}_t^c, \mathbf{b}_t^d} \quad \delta \sum_{\tau=t}^{t+h} \left[ \sum_{k=1}^{N_g} c_{k,\tau}^{\mathrm{g}} g_{k,\tau} + \sum_{k=1}^{N_b} c_{k,\tau}^{\mathrm{b,d}} b_{k,\tau}^{\mathrm{d}} - c_{k,\tau}^{\mathrm{b,c}} b_{k,\tau}^{\mathrm{c}} \right] \tag{4a}$$

$$\text{s.t.} \quad \Sigma [\mathbf{d}_\tau^\top, \mathbf{g}_\tau^\top, \mathbf{b}_\tau^{c\top}, \mathbf{b}_\tau^{d\top}]^\top = \mathbf{p}_\tau \qquad\qquad \forall \tau = t, \ldots, t+h \tag{4b}$$

$$\mathbf{1}^\top \mathbf{p}_\tau \delta = 0 \qquad\qquad \forall \tau = t, \ldots, t+h \tag{4c}$$

$$|H\mathbf{p}_t| \leq \mathbf{f}^{\max} \qquad\qquad \forall \tau = t, \ldots, t+h \tag{4d}$$

$$\mathbf{d}_\tau = \hat{\mathbf{d}}_\tau \qquad\qquad \forall \tau = t, \ldots, t+h \tag{4e}$$

$$\underline{\mathbf{g}} \leq \mathbf{g}_\tau \leq \overline{\mathbf{g}} \qquad\qquad \forall \tau = t, \ldots, t+h \tag{4f}$$

$$\mathbf{z}_\tau \in \{0,1\}^{N_b} \qquad\qquad \forall \tau = t, \ldots, t+h \tag{4g}$$

$$\mathbf{0} \leq \mathbf{b}_\tau^c \leq \overline{\mathbf{b}^c} \odot \mathbf{z}_\tau \qquad\qquad \forall \tau = t, \ldots, t+h \tag{4h}$$

$$\mathbf{0} \leq \mathbf{b}_\tau^d \leq \overline{\mathbf{b}^d} \odot (\mathbf{1} - \mathbf{z}_\tau) \qquad\qquad \forall \tau = t, \ldots, t+h \tag{4i}$$

$$\mathbf{x}_\tau = \mathbf{x}_{\tau-1} + \mathrm{diag}(\boldsymbol{\eta}^c)\mathbf{b}_\tau^c - \mathrm{diag}(\boldsymbol{\eta}^d)^{-1}\mathbf{b}_\tau^d \qquad\qquad \forall \tau = t, \ldots, t+h \tag{4j}$$

$$\underline{\mathbf{x}} \leq \mathbf{x}_\tau \leq \overline{\mathbf{x}} \qquad\qquad \forall \tau = t, \ldots, t+h \tag{4k}$$

$$\mathbf{x}_{t-1} = \mathbf{x}_{t-1}^* \tag{4l}$$

$$\mathbf{x}_{t+h} = \mathbf{x}^{\mathrm{f}} \tag{4m}$$

However, the mixed-integer linear program does not produce dual-variables the same way that the original linear program does. In order to recover nodal prices, we first solve the mixed-integer linear program to determine the optimal values for $\mathbf{z}_\tau$. Then, in the linear program, we replace $\overline{\mathbf{b}^c}$ with $\overline{\mathbf{b}^c} \odot \mathbf{z}_\tau$ and $\overline{\mathbf{b}^d}$ with $\overline{\mathbf{b}^d} \odot (\mathbf{1} - \mathbf{z}_\tau)$, and we solve the linear program to get the nodal prices from the dual variables. Although this procedure requires solving two optimization problems (first the MILP, and later the linear program), in practice solving the linear program is very fast, since the optimization variables can be initialized to their optimal values from the MILP.

**Market clearing and settlement.** The market clears when optimization problem (3) has a feasible solution, denoted $(\mathbf{g}_t^*, \mathbf{b}_t^{c*}, \mathbf{b}_t^{d*})$. The nodal vector of market clearing prices $\boldsymbol{\pi} \in \mathbb{R}^N$ (in \$/MWh) is defined by a function of dual variables $\lambda_t^*, \boldsymbol{\mu}_t^{+*}, \boldsymbol{\mu}_t^{-*}$:

$$\boldsymbol{\pi}_t := \lambda_t^* \mathbf{1} + H^\top (\boldsymbol{\mu}_t^{+*} - \boldsymbol{\mu}_t^{-*})$$

For each interval $t$, the settlement rule for generator $k$ at node $i$ is:

1. Generator $k$ produces power $g_{k,t}^*$
2. Generator $k$ receives revenue $\pi_{i,t} g_{k,t}^*$

The settlement rules for loads and batteries are analogous. When a battery is charging ($b_{k,t}^c > 0$), it pays the nodal price; when it is discharging ($b_{k,t}^d > 0$), it receives the nodal price.

**Representativeness and Generalizability** While `ElectricityMarketEnv` is not modeled after any particular real-world transmission network, it simulates the transmission network from the widely-used IEEE Reliability Test System (IEEE RTS-24) [22]. As the IEEE RTS-24 test case did not include electricity load data, we chose to incorporate load data from the recent 2019 IEEE RTS-GMLC update [23], which was designed to be representative of a modern transmission network located in the southwestern U.S., featuring a variety of renewable and distributed generators as well as representative electricity load profiles. According to industry experts, both the IEEE RTS-24 and IEEE RTS-GMLC are simplified but standard test cases. Furthermore, `ElectricityMarketEnv` has a modular design and is readily modified to simulate a particular network.

The multi-time-step security-constrained economic dispatch problem (SCED) implemented in `ElectricityMarketEnv` (3) is also representative of how market operators schedule generators and determine nodal prices in most electricity markets in the U.S.A.

**Limitations** Designing `ElectricityMarketEnv` to be easy to use necessitated some limitations in what could be modeled. First, the default IEEE RTS-24 network only features 24 buses, which

is smaller than most real-world transmission networks. However, because each step of the environment requires solving a mixed-integer linear program, a larger test case would have significantly increased the amount of time taken in the environment and thus slowed down any RL training. Users who wish to test their RL algorithms in larger transmission networks are welcome to modify `ElectricityMarketEnv` for their needs.

Second, `ElectricityMarketEnv` is only based on data representative of modern electricity markets in the U.S.A., whereas many regions around the world still feature vertically-integrated energy monopolies and/or "traditional" electricity markets that may not necessarily solve a similar SCED optimization problem.

Finally, `ElectricityMarketEnv` only features distribution shifts in the form of changes in the marginal carbon emissions and load profiles between summer and winter months. In the real-world, distribution shifts also occur when more generators are added to the network, when older generators are retired, and when the transmission network changes (*e.g.*, when new transmission lines are added upgraded, or when transmission lines are taken offline by extreme weather events).

### B.3   DatacenterEnv

**System Model and Motivation.**   `DatacenterEnv` is inspired by the carbon-aware job scheduling approach adopted by Google's datacenters, as described in [26]. The Google approach can be summarized as follows. Each day, the system plans the 24-hour virtual capacity curve (VCC) for the next day by solving a constrained optimization problem whose objective is to minimize a weighted sum of expected carbon emissions and expected peak power consumption. The main constraint is that the VCC for the next day must sum to the 97th-percentile of the predicted total daily capacity requirement. Capacity is measured in terms of normalized CPU compute units. The VCC artificially limits the total datacenter capacity at each hour, forcing enqueued jobs to be scheduled later than they would otherwise run. The goal is to reduce the number of running jobs when the carbon intensity of the electrical grid ($CO_2$ emissions per electrical energy used) is high, and run more jobs when the carbon intensity is low. By time-shifting jobs, the datacenter is able to reduce its $CO_2$ emissions.

The VCC-based approach to carbon-aware job scheduling is *scheduler-agnostic* as it works with any scheduler (*e.g.*, a FIFO queue, or a priority queue). The details of job placement (which physical machine runs each job) and job prioritization (which jobs run first) are up to the scheduler.

Whereas the Google approach plans a whole day's VCC at once, `DatacenterEnv` allows RL agents to plan the VCC one hour at a time. Furthermore, `DatacenterEnv` makes the following simplifying assumptions compared to the Google approach described above:

- `DatacenterEnv` does not provide any predictions of the next-day predicted total daily capacity, instead relying on the reward function to penalize an agent for failing to plan a VCC that meets 97% of the day's capacity.
- `DatacenterEnv` does not account for peak power consumption.
- `DatacenterEnv` uses a simple scheduler based on a priority queue, and jobs are randomly placed on any available machine. Machine constraints are not accounted for.

In `DatacenterEnv`, the data used to simulate a datacenter workload is subsampled from Cluster A from the Google Cluster Workload Traces May 2019 dataset [25]. The 47,276 jobs in our subsample includes widely varying priorities, ranging from 0 (low priority) to 450 (high priority, latency-sensitive). We assume that our datacenter consists of 101 identical machines.

**Training.**   We trained PPO, SAC, and deep deterministic policy gradient (DDPG) [33] RL algorithms on `DatacenterEnv` using PyTorch with RLLib's default hyperparameters for 60 episodes.

**Representativeness, Generalizability, and Limitations**   `DatacenterEnv` is loosely based on a Google data center from May 2019. As most datacenters do not disclose their exact job scheduling mechanisms and machine specifications, `DatacenterEnv` uses several simplifying assumptions. It uses a simple priority queue for scheduling jobs, and it assumes that there are no constraints for which jobs can be placed on which machine. However, `DatacenterEnv` does use a subsample of actual job traces from a Google datacenter in May 2019, which includes information for each job such as priority, duration, and compute usage. As `DatacenterEnv` is specifically based on the VCC framework used by Google's approach to carbon-aware datacenters, it does not reflect efforts by other

cloud computing providers such as Microsoft [34] that may incorporate carbon emission estimates directly into the decision-making of their datacenter job schedulers.

Because Google only released datacenter job traces from May 2019, our ability to provide distribution shifts in `DatacenterEnv` over time is limited. We have thus chosen to only test shifts in the marginal carbon emissions rate, even though more realistic distribution shifts would also include changes in the statistics of datacenter jobs and in the capacities of the datacenter machines.

### B.4 CogenEnv

**System Model and Motivation.** In the single-agent cogeneration environment, the agent controls a combined-cycle gas power plant with three gas turbines and one steam turbine. Gas power plants, similar to other conventional dispatchable generation types, suffer from loss of efficiency and degradation as a result of ramping energy generation up and down, thus necessitating the development of dispatch algorithms that can balance the tradeoff between fuel efficiency and ramp magnitude by anticipating future ramp needs while meeting demand in the highly constrained decision space. Such algorithms are even more crucial during the ongoing energy transition, as large quantities of variable and intermittent solar and wind resources are added to the grid. The variability of these resources requires dispatchable generators to ramp more frequently to balance supply with demand, and thus, dispatch algorithms that have been deployed historically may be suboptimal if they do not consider these ramp needs. The problem of dispatchable power plant operation in the face of uncertain renewable generation is thus an important problem, as better algorithms for generator dispatch will ensure that the performance and fuel efficiency of thermal generators will not degrade as renewables penetration increases.

The foundation of the system model in `CogenEnv` is a neural network that maps ambient conditions and dispatch variables to plant fuel consumption and other variables related to plant operation and constraints. This model was developed in partnership with Beyond Limits and Enexsa. A summary of the plant model inputs and outputs is provided in Table 5. In the table, "GT" abbreviates "Gas Turbine" and "HRSG" abbreviates "heat recovery steam generator". In brief, a generator dispatch decision includes generator outputs and steam flows for each of the three gas turbines, along with auxiliary binary decisions concerning the state of an evaporative cooler and power augmentation mode for the gas turbine. The dispatch decision also includes a generator output, condenser steam flow, and cooling tower bay count for the steam turbine. This dispatch decision, when provided to the plant model in conjunction with ambient temperature, pressure, and relative humidity (together comprising entries 1 through 18 in Table 5), yield a number of outputs (entries 19 through 47) detailing the fuel consumption of each turbine unit (entries 23-31) and of the plant as a whole (entry 22), the electric power generation of the plant (entry 19), the plant steam production (entry 20), and a number of dynamic operating limits on the dispatch variables (entries 32-47).

**Action Space.** The action space for the single-agent environment is a vector $a(t) \in \mathbb{R}^{15}$ specifying a dispatch decision for the cogeneration plant, and specifically, specifying values for entries 4 through 18 of Tables 5.

**Observation Space.** As described in the main body of the paper, observations take the form

$$s(t) = (\tau, a(t-1), T_{t:t+k}, P_{t:t+k}, H_{t:t+k}, d^p_{t:t+k}, d^q_{t:t+k}, \pi^p_{t:t+k}, \pi^f_{t:t+k}).$$

where $\tau$ is a normalized time – we consider each episode to be a single day, broken into 15 minute intervals, so $\tau = t/96$ – and $a(t-1)$ is the previous action. $T_{t:t+k}$, $P_{t:t+k}$, $H_{t:t+k}$, $d^p_{t:t+k}$, $d^q_{t:t+k}$, $\pi^p_{t:t+k}$ and $\pi^f_{t:t+k}$ are vectors of the current and $k$ steps of future forecasts of temperature, pressure, relative humidity, electricity demand, steam demand, electricity price, and fuel price, respectively. The units and limits of temperature, pressure, and relative humidity are as in entries 1 through 3 of Table 5, electricity demand has units MW, and steam demand is in klb/h. While we do not use electricity and fuel prices in the reward for this environment, we include them in the observation to allow for modification of the agent reward to incorporate financial incentives; gas price data was obtained from the Henry Hub Natural Gas Spot Price dataset (`https://www.eia.gov/dnav/ng/hist/rngwhhdm.htm`) and electricity price data was obtained from historical day-ahead prices at the Houston zone of the ERCOT grid (`https://www.ercot.com/mp/data-products/data-product-details?id=NP4-180-ER`).

Table 5: Summary of input and output variables for `CogenEnv` plant model

| Num. | Variable Description | Variable Group | Type | Feasible Region | Unit |
|------|---------------------|----------------|------|-----------------|------|
| 1 | Air Temperature | Ambient Conditions | Input | $[32, 115]$ | °F |
| 2 | Air Pressure | Ambient Conditions | Input | $[14, 15]$ | PSIA |
| 3 | Air Relative Humidity | Ambient Conditions | Input | $[0, 1]$ | - |
| 4 | GT1 Generator Output | Turbine Block 1 | Input | $[41.64, 168.3]$ | MW |
| 5 | GT1 Evaporative Cooler Switch | Turbine Block 1 | Input | $\{0, 1\}$ | - |
| 6 | GT1 Power Augmentation Switch | Turbine Block 1 | Input | $\{0, 1\}$ | - |
| 7 | GT1 Steam Flow | Turbine Block 1 | Input | $[403.2, 819.6]$ | klb/h |
| 8 | GT2 Generator Output | Turbine Block 2 | Input | $[41.49, 168.4]$ | MW |
| 9 | GT2 Evaporative Cooler Switch | Turbine Block 2 | Input | $\{0, 1\}$ | - |
| 10 | GT2 Power Augmentation Switch | Turbine Block 2 | Input | $\{0, 1\}$ | - |
| 11 | GT2 Steam Flow | Turbine Block 2 | Input | $[396.7, 817.4]$ | klb/h |
| 12 | GT3 Generator Output | Turbine Block 3 | Input | $[46.46, 172.4]$ | MW |
| 13 | GT3 Evaporative Cooler Switch | Turbine Block 3 | Input | $\{0, 1\}$ | - |
| 14 | GT3 Power Augmentation Switch | Turbine Block 3 | Input | $\{0, 1\}$ | - |
| 15 | GT3 Steam Flow | Turbine Block 3 | Input | $[439.0, 870.3]$ | klb/h |
| 16 | Steam Generator Output | Steam Turbine | Input | $[25.65, 83.54]$ | MW |
| 17 | Steam Flow through Condenser | Steam Turbine | Input | $[-1218, -318.1]$ | klb/h |
| 18 | Number of Cooling Tower Bays | Steam Turbine | Input | $\{1, \ldots, 12\}$ | - |
| 19 | Net Electric Power Output | Plant | Output | - | MW |
| 20 | Net Steam Export | Plant | Output | - | klb/h |
| 21 | Auxiliary Power Consumption | Plant | Output | - | MW |
| 22 | Total Fuel Consumption | Plant | Output | - | klb/h |
| 23 | GT1 Fuel Flow | Turbine Block 1 | Output | - | klb/h |
| 24 | HRSG1 Fuel Flow | Turbine Block 1 | Output | - | klb/h |
| 25 | Block 1 Total Fuel Consumption | Turbine Block 1 | Output | - | klb/h |
| 26 | GT2 Fuel Flow | Turbine Block 2 | Output | - | klb/h |
| 27 | HRSG2 Fuel Flow | Turbine Block 2 | Output | - | klb/h |
| 28 | Block 2 Total Fuel Consumption | Turbine Block 2 | Output | - | klb/h |
| 29 | GT3 Fuel Flow | Turbine Block 3 | Output | - | klb/h |
| 30 | HRSG3 Fuel Flow | Turbine Block 3 | Output | - | klb/h |
| 31 | Block 3 Total Fuel Consumption | Turbine Block 3 | Output | - | klb/h |
| 32 | GT1 Min Power | Turbine Block 1 | Output | - | MW |
| 33 | GT1 Max Power | Turbine Block 1 | Output | - | MW |
| 34 | GT2 Min Power | Turbine Block 2 | Output | - | MW |
| 35 | GT2 Max Power | Turbine Block 2 | Output | - | MW |
| 36 | GT3 Min Power | Turbine Block 3 | Output | - | MW |
| 37 | GT3 Max Power | Turbine Block 3 | Output | - | MW |
| 38 | GT1 Min Steam | Turbine Block 1 | Output | - | klb/h |
| 39 | GT1 Max Steam | Turbine Block 1 | Output | - | klb/h |
| 40 | GT2 Min Steam | Turbine Block 2 | Output | - | klb/h |
| 41 | GT2 Max Steam | Turbine Block 2 | Output | - | klb/h |
| 42 | GT3 Min Steam | Turbine Block 3 | Output | - | klb/h |
| 43 | GT3 Max Steam | Turbine Block 3 | Output | - | klb/h |
| 44 | Steam Let-Down Flow Min Limit | Steam Turbine | Output | - | klb/h |
| 45 | Steam Let-Down Flow Max Limit | Steam Turbine | Output | - | klb/h |
| 46 | Steam Turbine Min Power | Steam Turbine | Output | - | MW |
| 47 | Steam Turbine Max Power | Steam Turbine | Output | - | MW |

**Reward Function** The reward for the agent is defined as

$$r(t) = - \left( r_f(a(t); T_t, P_t, H_t) + r_r(a(t); a(t-1)) + r_c(a(t); d_t^p, d_t^q) \right).$$

The term $r_f(a(t); T_t, P_t, H_t)$ is the generator fuel consumption in response to dispatch decision $a(t)$; this is exactly the total fuel consumption of the plant (entry 22 of Table 5) resulting from model inputs $(T_t, P_t, H_t, a(t))$. The term $r_r(a(t); a(t-1))$ is the cost of ramping electricity generation up or down. Defining $a_j(t)$ to be the $j$th entry of Table 5 (so $j$ ranges from 4 to 18 to include all entries comprising the action space), the ramp cost is defined as

$$r_r(a(t); a(t-1)) = \beta \cdot (|a_4(t) - a_4(t-1)| + |a_8(t) - a_8(t-1)| \\ + |a_{12}(t) - a_{12}(t-1)| + |a_{16}(t) - a_{16}(t-1)|).$$

In our experiments, we set the penalty magnitude $\beta = 2$, following [35]. The third term, $r_c(a(t); d_t^p, d_t^q)$, penalizes two types of constraint violation: the first is supply-demand imbalance. Let $x(t) \in \mathbb{R}^{47}$ be a vector containing plant model inputs and outputs as in Table 5; then $x_{19}(t)$ is the total power output of the plant and $x_{20}(t)$ is the total steam output of the plant. The form of the penalty on supply-demand imbalance is as follows:

$$r_c^{\mathrm{sd}}(a(t); d_t^p, d_t^q) = \gamma \cdot \left( \max\{0, d_t^p - x_{19}(t)\} + \max\{0, d_t^q - x_{20}(t)\} \right).$$

The second type of constraint violation penalized is that of the dynamic operating constraints. These are determined by the plant outputs in entries 32 through 47 of Table 5, and constrain dispatch variables to lie within certain intervals. This penalty takes the following form:

$$
\begin{aligned}
r_c^{\mathrm{dyn}}(a(t); d_t^p, d_t^q) = \gamma \cdot \big( & \max\{0, x_4(t) - x_{32}(t)\} + \max\{0, x_{33}(t) - x_4(t)\} \\
& + \max\{0, x_8(t) - x_{34}(t)\} + \max\{0, x_{35}(t) - x_8(t)\} \\
& + \max\{0, x_{12}(t) - x_{36}(t) + \max\{0, x_{37}(t) - x_{12}(t)\} \\
& + \max\{0, x_7(t) - x_{38}(t)\} + \max\{0, x_{39}(t) - x_7(t)\} \\
& + \max\{0, x_{11}(t) - x_{40}(t)\} + \max\{0, x_{41}(t) - x_{11}(t)\} \\
& + \max\{0, x_{15}(t) - x_{42}(t)\} + \max\{0, x_{43}(t) - x_{15}(t)\} \\
& + \max\{0, x_{17}(t) - x_{44}(t)\} + \max\{0, x_{17}(t) - x_{45}(t)\} \\
& + \max\{0, x_{16}(t) - x_{46}(t)\} + \max\{0, x_{47}(t) - x_{16}(t)\} \big).
\end{aligned}
$$

In our experiments, we set the penalty magnitude parameter $\gamma = 1000$. The total constraint violation penalty is simply the sum of the supply-demand violation penalty $r_c^{\mathrm{sd}}(a(t); d_t^p, d_t^q)$ and the dynamic operating constraint penalty $r_c^{\mathrm{dyn}}(a(t); d_t^p, d_t^q)$.

**Distribution Shift.** In our distribution shift scenario, we consider the addition of 300 MW of wind energy onto the grid, which increases variability of net energy demand and changes the shape of load profiles. We obtain simulated wind speed profiles using the WIND Toolkit [36] at an altitude of 100m at $(39.970406, -128.77481)$ at 15 minute intervals, and transform these into electricity generation profiles using the power curve for an IEC Class 2 turbine, scaling to obtain a maximum generation level of 300 MW.

**Multi-Agent Setting.** In the multi-agent version of the environment, each of the four agents represents one of the blocks of the cogeneration plant: the first three control the three gas turbine blocks, and the fourth controls the steam generation block. Each agent observes the global observation, but controls only the variables relevant for its block — thus, agent 1 controls variables 4 through 7 in Table 5, agent 2 controls variables 8 through 11, agent 3 controls variables 12 through 15, and agent 4 controls variables 16 through 18. Each agent's reward only includes the terms relevant for its unit - thus, for instance, agent 1 pays fuel cost according to entry 25, ramp cost $\beta \cdot (|a_4(t) - a_4(t-1)|)$, and dynamic operating constraint penalty $\gamma \cdot \big( \max\{0, x_4(t) - x_{32}(t)\} + \max\{0, x_{33}(t) - x_4(t)\} + \max\{0, x_7(t) - x_{38}(t)\} + \max\{0, x_{39}(t) - x_7(t)\} \big)$. However, the supply-demand imbalance constraint penalty is shared equally amongst all agents: each pays $\frac{1}{4} r_c^{\mathrm{sd}}(a(t); d_t^p, d_t^q)$.

**Training.** We train and test the performance of several algorithms on 250 days of data (ambient conditions, demands, and prices) between May 2021 and January 2022. For the testing environment, we use the scenario with 300 MW of wind generation. We performed two splits: 1) by algorithm (PPO vs. random actions), and 2) training data with out-of-distribution data (300 MW wind generation) vs. in-distribution data (no wind generation). We tested 3 learning rates for PPO and multi-agent PPO: 5e-6, 5e-5, 5e-4. For each learning rate, three different random seeds were tested, and the model with the best training performance was selected.

**Representativeness, Generalizability, and Limitations** CogenEnv is based on an actual combined-cycle power plant operated in the U.S.A. The general configuration has remained unchanged, but certain parameters such as generation capacity of units have been changed to protect the original data. More specifically, the general configuration of gas turbines, steam turbine, cooling tower, and binary setpoints are the same. Temperature and pressure settings of intermediate pressure and high pressure steams are also the same. However, some design parameters such as power and steam generation capacities of each unit and units' efficiencies has been scaled/shifted.

According to our collaborator Mehdi Hosseini at Beyond Limits: "Gas and steam turbines are generally modeled by simulating the Brayton and Rankine thermodynamic cycles. Besides the considered setpoints, some gas turbines might include extra minor setpoints like switching the duct burner on and off, which is considered always on in this model. The behavior of the steam turbine is more complex and interrelated with the structure of the condenser and cooling tower. In this simulation, a forced air cooling tower structure is employed, a typical method in combined cycle power plants. It's worth noting that using a different cooling system might influence the efficiency and generation limits of the steam turbine. In summary, the model accurately represents single or multi-gas turbine power plants, as well as combined-cycle power plants with a forced air cooling tower."

Finally, the distribution shift modeled in `CogenEnv` comes from changes in external renewable wind energy penetration, which causes greater need for ramping of fuel-based generators, leading to higher ramping costs. The is reflective of actual changes and challenges occurring in modern electricity grids, often referred to as the "duck curve problem" [37]. However, `CogenEnv` currently does not model other sources of distribution shift such as changing fuel prices.

## B.5 BuildingEnv

Table 6: Definitions of `BuildingEnv` variables.

| Variable | Domain | Unit | Description |
|---|---|---|---|
| $T_i(t)$ | $\mathbb{R}$ | °C | temperature in zone $i$ at time $t$ |
| $T_G(t)$ | $\mathbb{R}$ | °C | ground temperature at time $t$ |
| $T_E(t)$ | $\mathbb{R}$ | °C | outdoor/ambient temperature at time $t$ |
| $Q^{\mathrm{GHI}}(t)$ | $\mathbb{R}_+$ | W/m² | heat gain from solar irradiance per square meter at time $t$ ("GHI" is an acronym for "global horizontal irradiance") |
| $Q_i^{\mathrm{s}}(t)$ | $\mathbb{R}_+$ | W | heat acquisition from solar irradiance in zone $i$ at time $t$ |
| $Q_i^{\mathrm{p}}(t)$ | $\mathbb{R}_+$ | W | heat acquisition from occupants' activities in zone $i$ at time $t$ |
| $Q_i^{\mathrm{h}}(t)$ | $\mathbb{R}$ | W | heat acquisition from HVAC in zone $i$ at time $t$ |
| $N_i(t)$ | $\mathbb{N}$ | | number of occupants in zone $i$ at time $t$ |
| $A_i^{\mathrm{win}}$ | $\mathbb{R}_+$ | m² | window area for zone $i$ |
| $Q_i^{\mathrm{h,max}}$ | $\mathbb{R}_+$ | W | maximum heat acquisition from HVAC in zone $i$ |
| $a_i(t)$ | $[-1, 1]$ | | action value, normalized heat acquisition from HVAC in zone $i$ at time $t$ (+ for heating, − for cooling) |
| $w_i$ | $[0, 1]$ | | efficiency coefficient for HVAC in zone $i$ |

**Observation space.** `BuildingEnv` considers a building with $M$ indoor zones. The observation $s(t) \in \mathbb{R}^{M+4}$ is the concatenation of the zonal observations $T(t) \in \mathbb{R}^M$ and the environmental observations $s^{\mathrm{env}}(t) \in \mathbb{R}^4$:

$$s(t) = [T(t)^\top, s^{\mathrm{env}}(t)^\top]^\top$$
$$T(t) = [T_1(t), \dots, T_M(t)]^\top$$
$$s^{\mathrm{env}}(t) = [T_E(t), T_G(t), Q^{\mathrm{GHI}}(t), \bar{Q}^{\mathrm{p}}(t)]^\top.$$

We split the observation $s(t)$ into the two components to emphasize that $T(t)$ is affected by the agent's control actions, whereas $s^{\mathrm{env}}(t)$ is the set of exogenous time-varying environmental variables that are unaffected by the agent's control actions. The descriptions for each of these variables can be found in Table 6.

**Action space.** The action $a(t) = [a_1(t), \dots, a_M(t)]^\top \in [-1, 1]^M$ is a set of controllable actions for building heat control, where $a_i(t)$ is the controlled heating supplied to zone $i$. We normalize the HVAC power consumption with respect to the maximum heating capacity, so that $a_i(t)$ is bounded in $[-1, 1]$. The resulting heat supply from the HVAC system to zone $i$ at time $t$ is given by

$$Q_i^{\mathrm{h}}(t) = a_i w_i Q_i^{\mathrm{h,max}},$$

where $w_i$ is the efficiency coefficient for the HVAC system in zone $i$ and $Q_i^{\mathrm{h,max}}$ is the maximum heat supply from the HVAC in zone $i$.

Given an action $a(t)$, the `BuildingEnv.step()` method simulates the next state via the physics-based state transition model described below.

**System model.** The building simulation builds upon the physics-principled environment in [38], which includes a reduced linear Resistance-Capacitance (RC) model for heat transfer with nonlinear residual modeling for occupants' activities and solar irradiance. The zonal thermal dynamics are given by

$$C_i \frac{dT_i}{dt} = \sum_{j \in \mathcal{N}(i)} \frac{T_j - T_i}{R_{i,j}} + Q_i^{\mathrm{h}} + Q_i^{\mathrm{s}} + Q_i^{\mathrm{p}}, \tag{5}$$

where $\mathcal{N}(i)$ is the set of zones neighboring zone $i$. That is,

$$\mathcal{N}(i) = \{j \in \{1, \ldots, M, \mathrm{G}, \mathrm{E}\} \mid j \neq i, \text{ zone } j \text{ shares a wall with zone } i\}.$$

The set $\mathcal{N}(i)$ may include G and E, if zone $i$ is on the ground floor or connected to the outside environment, respectively. $C_i$ is the thermal capacitance (in J/K), and $R_{i,j} = R_{j,i}$ is the thermal resistance (in K/W) between zones $i, j$. If zone $i$ and $j$ are not neighboring zones (*i.e.*, $j \notin \mathcal{N}(i)$), we set $R_{i,j} = R_{j,i} = +\infty$. $Q_i^{\mathrm{h}}$ is the controlled heating/cooling flow distributed to each zone as described above. $Q_i^{\mathrm{s}}$ and $Q_i^{\mathrm{p}}$ adhere to models outlined in the EnergyPlus documentation [39], which represent the heat accumulation in zone $i$ from solar heat acquisition from windows and indoor human activities, respectively.

To capture the solar heat acquisition from windows, let $\alpha_{\mathrm{GHI}}$ denote the coefficient determining the solar heat gain for windows, $A_i^{\mathrm{win}}$ be the window area for zone $i$ (in m$^2$), and $Q^{\mathrm{GHI}}(t)$ be the heat gain from global horizontal irradiance (in W/m$^2$). Then the accumulated solar heat from windows at time $t$ can be calculated as

$$Q_i^{\mathrm{s}}(t) = \alpha_{\mathrm{GHI}} \cdot A_i^{\mathrm{win}} \cdot Q^{\mathrm{GHI}}(t).$$

To calculate heat gain originating from human activities at time $t$, $Q_i^{\mathrm{p}}(t)$, we consider $N_i$ to symbolize the population in zone $i$, and $\bar{Q}^{\mathrm{p}}(t)$ signifies the discernible heat contributed by a single individual's activities. We assume that the number of people in each zone does not change over the course of the simulation, and we leave the task of modeling time-varying population for future work. Then,

$$Q_i^{\mathrm{p}}(t) = \bar{Q}^{\mathrm{p}}(t) \cdot N_i.$$

The computation of sensible heat per individual denoted as $\bar{Q}_p(t)$, is given by a polynomial function from the EnergyPlus documentation [39, p.1299],

$$\bar{Q}^{\mathrm{p}}(t) = c_1 + c_2 m_t + c_3 m_t^2 + c_4 \bar{T}(t) - c_5 m_t \bar{T}(t) + c_6 m_t^2 \bar{T}(t) - c_7 \bar{T}^2(t) + c_8 m_t \bar{T}^2(t) - c_9 m_t^2 \bar{T}^2(t),$$

where $m_t$ signifies the population metabolic rate (in W) at time $t$, $\bar{T}(t) = \frac{1}{M} \sum_{i=1}^{M} T_i(t)$ is the average zone temperature, and $c_1, \ldots, c_9$ are constants that have been deduced by fitting sensible heat data under a variety of conditions. Both $Q_i^{\mathrm{s}}(t)$ and $Q_i^{\mathrm{p}}(t)$ fluctuate over time and embody heat emanations from the environment and occupants' activities that are beyond control.

The state evolution in (5), together with the definitions of $Q_i^{\mathrm{h}}, Q_i^{\mathrm{s}}, Q_i^{\mathrm{p}}$ can be written as

$$\dot{T}(t) = AT(t) + Bu(t) + Df(T(t)), \tag{6}$$

where $u(t) = [T_{\mathrm{G}}(t), T_{\mathrm{E}}(t), a_1(t), \ldots, a_M(t), Q^{\mathrm{GHI}}(t)]^\top$. Let the indicator variables $\mathbf{1}_{\mathrm{G},i} := \mathbf{1}[\mathrm{G} \in \mathcal{N}(i)]$ and $\mathbf{1}_{\mathrm{E},i} := \mathbf{1}[\mathrm{E} \in \mathcal{N}(i)]$ encode zone $i$'s connectivity to the ground and the outside environment, respectively. Then the $A$ and $B$ matrices are

$$A = \begin{bmatrix} \sum_{j \in \mathcal{N}(i)} \frac{-1}{C_1 R_{1,j}} + \frac{c_p N_1}{MC_1} & \frac{1}{C_1 R_{1,2}} + \frac{c_p N_1}{MC_1} & \cdots & \frac{1}{C_1 R_{1,M}} + \frac{c_p N_1}{MC_1} \\ \frac{1}{C_2 R_{2,1}} + \frac{c_p N_2}{MC_2} & \sum_{j \in \mathcal{N}(i)} \frac{-1}{C_2 R_{2,j}} + \frac{c_p N_2}{MC_2} & \ddots & \vdots \\ \vdots & & \ddots & \frac{1}{C_{M-1} R_{M-1,M}} + \frac{c_p N_{M-1}}{MC_{M-1}} \\ \frac{1}{C_M R_{M,1}} + \frac{c_p N_M}{MC_M} & \cdots & \cdots & \sum_{j \in \mathcal{N}(i)} \frac{-1}{C_M R_{M,j}} + \frac{c_p N_M}{MC_M} \end{bmatrix},$$

$$B = \begin{bmatrix} \frac{\mathbf{1}_{\mathrm{G},1}}{C_1 R_{\mathrm{G},1}} & \frac{\mathbf{1}_{\mathrm{E},1}}{C_1 R_{\mathrm{E},1}} & \frac{w_1 Q_1^{\mathrm{h,max}}}{C_1} & 0 & \cdots & 0 & \frac{\alpha_{\mathrm{GHI}} A_1^{\mathrm{win}}}{C_1} \\ \frac{\mathbf{1}_{\mathrm{G},2}}{C_2 R_{\mathrm{G},2}} & \frac{\mathbf{1}_{\mathrm{E},2}}{C_2 R_{\mathrm{E},2}} & 0 & \frac{w_2 Q_2^{\mathrm{h,max}}}{C_2} & \cdots & 0 & \frac{\alpha_{\mathrm{GHI}} A_2^{\mathrm{win}}}{C_2} \\ \vdots & \vdots & \vdots & & \ddots & \vdots & \vdots \\ \frac{\mathbf{1}_{\mathrm{G},M}}{C_M R_{\mathrm{G},M}} & \frac{\mathbf{1}_{\mathrm{E},M}}{C_M R_{\mathrm{E},M}} & 0 & 0 & \cdots & \frac{w_M Q_M^{\mathrm{h,max}}}{C_M} & \frac{\alpha_{\mathrm{GHI}} A_M^{\mathrm{win}}}{C_M} \end{bmatrix}.$$

The vector $D$ in Eq (6) is defined as $D = \left[\frac{N_1}{C_1}, \frac{N_2}{C_2}, \cdots, \frac{N_M}{C_M}\right]^\top$, and the nonlinear function for sensible heat calculation is

$$
\begin{aligned}
f(T(t)) &:= \bar{Q}^{\mathrm{p}}(t) - c_4 \bar{T}(t) \\
&= c_1 + c_2 m_t + c_3 m_t^2 - c_5 m_t \bar{T}(t) + c_6 m_t^2 \bar{T}(t) - c_7 \bar{T}^2(t) + c_8 m_t \bar{T}^2(t) - c_9 m_t^2 \bar{T}^2(t).
\end{aligned}
$$

We convert the continuous-time system model into a discrete-time model,

$$
T[k+1] = \bar{A} T[k] + \bar{B} u[k] + D \bar{f}[k] \tag{7}
$$

where the discrete-time system matrices $\bar{A}, \bar{B}, \bar{f}$ are obtained from the continuous-time model parameters $A, B, f(T(t))$ using zero-order hold method with discretization time $\Delta T$. In particular, $\bar{A} = e^{A \Delta T}$, $\bar{B} = A^{-1}(\bar{A} - I) B$, and $\bar{f}[k] = \int_{k\Delta T}^{(k+1)\Delta T} f(T(\tau)) \, \mathrm{d}\tau$.

**Reward function.** The main objective of building control is to reduce energy consumption while keeping the temperature within a given comfort range. Therefore, the reward function penalizes both temperature deviations and HVAC energy consumption:

$$
r(t) = -(1 - \beta) \left\| a(t) \right\|_p - \beta \| T^{\mathrm{target}}(t) - T(t) \|_p \,,
$$

where $T^{\mathrm{target}}(t) = [T_1^{\mathrm{target}}(t), \ldots, T_M^{\mathrm{target}}(t)]^\top$ are the target temperatures. The parameters $\beta$ and $p$ are user-customizable scalars, where $\beta$ trades off between the energy consumption and temperature deviation penalties, and $p$ determines the norm used. An important future direction is to incorporate $CO_2$ emissions into consideration in the default reward function for building control environment.

**Building and weather types for simulation.** The prototype buildings included in `BuildingEnv` are derived from the Department of Energy (DOE) Commercial Reference Building Models. The models include 16 commercial building types in 19 locations. Users can download all models at `https://www.energycodes.gov/prototype-building-models`. Since `BuildingEnv` is compatible with EnergyPlus, users could also create their own model in the EnergyPlus editor and load the generated table file into the `BuildingEnv`.

- *Available building types*: ApartmentHighRise, ApartmentMidRise, Hospital, HotelLarge, HotelSmall, OfficeLarge, OfficeMedium, OfficeSmall, OutPatientHealthCare, RestaurantFastFood, RestaurantSitDown, RetailStandalone, RetailStripmall, SchoolPrimary, SchoolSecondar, Warehouse.

- *Available cities and weather types*: Ho Chi Minh City (Extremely Hot Humid), Dubai (Extremely Hot Dry), Honolulu (Very Hot Humid), New Delhi (Very Hot Dry), Tampa (Hot Humid), Tucson (Hot Dry), Atlanta (Warm Humid), El Paso (Warm Dry), San Diego (Warm Marine), New York (Mixed Humid), Albuquerque (Mixed Dry), Seattle (Mixed Marine), Buffalo (Cool Humid), Denver (Cool Dry), Port Angeles (Cool Marine), Rochester (Cold Humid), Great Falls (Cold Dry), International Falls (Very Cold), Fairbanks (Subarctic/Arctic).

**Example.** A practical usage example is demonstrated in Figure 8. The intent of this code excerpt is to imitate an "OfficeLarge" type structure in San Diego, with "Warm Marine" weather conditions, employing arbitrarily selected actions. Upon the creation of a building model, comprehensive zone information is then displayed as shown in Figure 8. At each control time step, the RL agent observes the present state $s(t)$ and produces a corresponding action $a(t)$. The environment `BuildingEnv`, in turn, incorporates this action and integrates it into the building state-space model to project the subsequent state $s(t+1)$ for the approaching time step. Each episode runs for 1 day, with 5-minute time intervals ($H = 288$, $\tau = 5/60$ hours). The agent controls the supplied heat flow to each zone and is rewarded for maintaining the desired temperature at the minimum electricity usage. Figure 9 visualizes the indoor temperature in different zones of OfficeLarge for 1 day without control, initialized at $13.2^\circ C$.

**Model predictive control (MPC).** As a baseline non-RL algorithm, we consider a model predictive control (MPC) controller similar to the `EVChargingEnv` environment. At every time step $t$, with a

```
>>> from sustaingym.envs.building import BuildingEnv, ParameterGenerator
>>> params = ParameterGenerator(building='OfficeLarge', weather='Warm_Marine', location='SanDiego')
>>> env = BuildingEnv(params)

###############All Zones from Ground############
BASEMENT  [Zone index]:  0
DATACENTER_BASEMENT_ZN_6  [Zone index]:  1
CORE_BOTTOM  [Zone index]:  2
PERIMETER_BOT_ZN_3  [Zone index]:  3
PERIMETER_BOT_ZN_2  [Zone index]:  4
PERIMETER_BOT_ZN_1  [Zone index]:  5
PERIMETER_BOT_ZN_4  [Zone index]:  6
DATACENTER_BOT_ZN_6  [Zone index]:  7
GROUNDFLOOR_PLENUM  [Zone index]:  8
CORE_MID  [Zone index]:  9
PERIMETER_MID_ZN_3  [Zone index]:  10
PERIMETER_MID_ZN_2  [Zone index]:  11
PERIMETER_MID_ZN_1  [Zone index]:  12
PERIMETER_MID_ZN_4  [Zone index]:  13
DATACENTER_MID_ZN_6  [Zone index]:  14
MIDFLOOR_PLENUM  [Zone index]:  15
CORE_TOP  [Zone index]:  16
PERIMETER_TOP_ZN_3  [Zone index]:  17
PERIMETER_TOP_ZN_2  [Zone index]:  18
PERIMETER_TOP_ZN_1  [Zone index]:  19
PERIMETER_TOP_ZN_4  [Zone index]:  20
DATACENTER_TOP_ZN_6  [Zone index]:  21
TOPFLOOR_PLENUM  [Zone index]:  22
#################################################
```

Figure 8: Importing OfficeLarge in `BuildingEnv`.

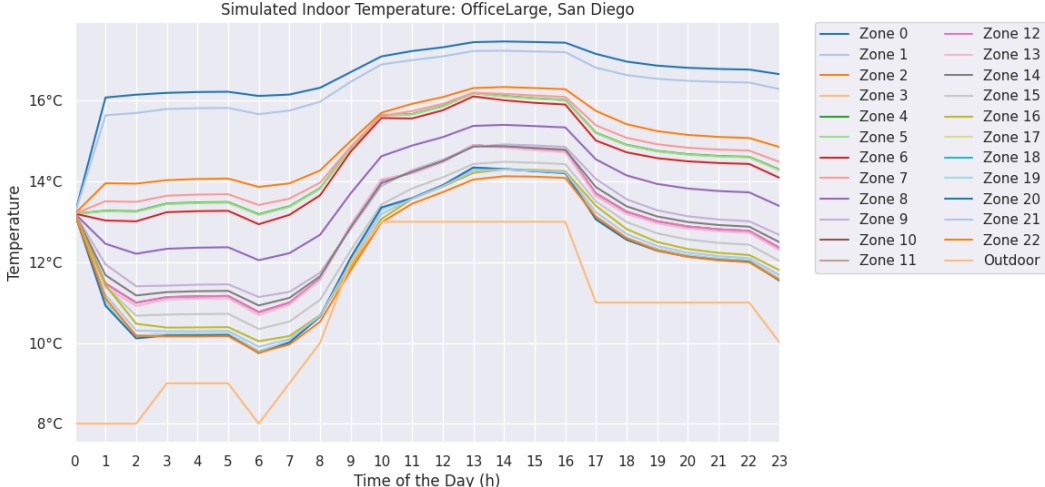

Figure 9: Simulated indoor temperature in different zones of OfficeLarge for 1 day without control.

lookahead window $w$, the MPC controller solves the following optimization problem,

$$\min_{a(t),\ldots,a(t+w-1)} \sum_{k=t}^{t+w-1} (1-\beta)\,\|a(k)\|_2 + \beta\|T^{\text{target}}(k) - T(k)\|_2 \tag{8a}$$

$$\text{s.t.} \quad -1 \preceq a_i(k) \preceq 1\,, \forall i \in \{0,\ldots,M\}, \forall k \in \{t,\ldots,t+w-1\} \tag{8b}$$

$$\text{building dynamics in (7)}\,, \tag{8c}$$

and implements action $a^\star(t)$. MPC method has complete model knowledge and perfect predictions about building occupancy $[N_1, N_2, \ldots, N_M]$, the ground and environmental temperature $T_G(k), T_E(k)$, solar irradiance $Q^{\text{GHI}}(k)$ for $k = t, t+1, \ldots, t+w-1$. (8b) ensures that the control signals are feasible, and (8c) follows the building physics model in (7) which uses a zero-order hold discretization method to derive from the continuous-time model with discretization interval as $\Delta T = 1/12$ hour.

**Multiagent Environment.** The multiagent setting pairs each zone with one agent, so that each agent $i$ is only responsible for action $a_i(t)$. Currently, we let every agent observe the complete building state space, and each agent receives the same global reward. For future directions, we plan

on implementing a separate local state space for each agent and also imposing a total power constraint (*e.g.*, $\|a(t)\|_1$ must be bounded by some limit).

**Training**    We trained Proximal Policy Optimization (PPO), Soft Actor-Critic (SAC), and Advantage Actor Critic (A2C) [40] Reinforcement Learning algorithms on the `BuildingEnv` environment using PyTorch and StableBaselines3. These algorithms were applied to an "OfficeSmall" type building situated in a "Hot Dry" climate, specifically in Tucson. The training and testing periods were set in the winter (January 2003) and summer (June 2004) sections, respectively. The models were trained during both the winter and summer sections, but testing was conducted solely in the winter section to assess the impact of distribution shifts. We experimented with two learning rates for each of PPO, SAC, and A2C: 3e-4 and 3e-5 for PPO and SAC, and 7e-4 and 3e-4 for A2C. The training used a time resolution of 300 seconds (or 5 minutes). The model demonstrating the best performance in training was subsequently selected for further analysis.

**Representativeness and Generalizability**    `BuildingEnv` is designed to ensure broad generalizability across diverse building scenarios. We have integrated an extensive collection of standard prototype building models coupled with a variety of weather conditions, encompassing the majority of Reference Building types. These models are based on the Department of Energy (DOE)'s Commercial Reference Building Models, offering a robust foundation for simulation. Furthermore, the environment is highly customizable. Users have the flexibility to adjust parameters such as the wall material, the solar heat gain coefficient of windows, and the ground temperature specific to the building's location. For researchers and developers seeking even more specificity, there is also an option to define custom building models using EnergyPlus. Users can then input the resulting output input data file (IDF), along with the corresponding EPW weather file, into `BuildingEnv`, providing a custom simulation experience.

**Limitations**    Beyond the modeled weather variations, occupancy dynamics also introduce distribution shifts in building energy management. While `BuildingEnv` provides mechanisms to simulate occupants in different zones with set activity schedules, truly replicating the unpredictability of human behavior remains a complex endeavor. For instance, we can configure a room to have four individuals engaged in seated work from 1pm to 4pm, followed by three individuals running from 4pm to 5pm, with the room being vacant after 5pm. Despite these features, there are certain distribution shifts that `BuildingEnv` does not currently address, such as equipment failures, external environmental impacts like nearby construction or urban heat island effects, and the nuances of building aging. However, it is worth noting that the modular design of `BuildingEnv` provides a foundation that is conducive to future enhancements and adaptations, allowing for the incorporation of additional sources of distribution shift as our understanding of building dynamics evolves.