# OpenReview forum: "SustainGym: Reinforcement Learning Environments for Sustainable Energy Systems"
_NeurIPS.cc/2023/Track/Datasets_and_Benchmarks — NeurIPS 2023 Datasets and Benchmarks Poster_

### Official Review · Reviewer_E4iv · 2023-07-03
**A good idea and useful contribution to the community, but needs more work.**

**Rating:** 5
**Confidence:** 4
**Correctness:** The methodology and experimental resu…

**Strengths:**

- The development of SustainGym addresses the need for more realistic and sustainable RL environments that model real-world scenarios.
- The environments test RL algorithms under distribution shifts and multi-agent settings, which can provide researchers with a better understanding of how algorithms would perform in real-world situations.
- The paper offers an extensive evaluation of current off-the-shelf RL algorithms using SustainGym, which is beneficial for future research and benchmarking.

**Additional Feedback:**

SustainGym is a great idea and I definitely think this work will be publishable once some of the above items have been addressed, but its current state is too under-developed for me to recommend its acceptance at this time.

**Clarity:**

The paper is well-written, clear, and organized, providing a detailed explanation of the SustainGym environments. However, it could be improved by a more detailed discussion of multi-agent settings and physical constraints within the main body of the paper, rather than in the appendix. Discussion of the limitations of SustainGym and the representativeness of its environments is also required.

**Documentation:**

The authors have provided sufficient documentation with the paper, including a GitHub repository that contains the code, licenses, and usage instructions for SustainGym.

**Ethics:**

The paper seems to follow ethical guidelines, focusing on promoting the application of RL to sustainability tasks, which could contribute to societal and environmental benefits. There are no apparent ethical concerns.

**Limitations:**

The discussion is very brief and I would expect to see some acknowledgment of the limitations of SustainGym. An obvious limitation is the lack of consideration of other sustainability dimensions. Sustainability goes beyond considerations of just energy and CO2 emissions and should consider factors such as water usage, eutrophication potential, etc., where possible. Another limitation is the representativeness of these sustainable energy tasks.

**Opportunities For Improvement:**

- While the authors provide a useful introduction to the five environments, they could offer more justifications for their design decisions.
- The experiments conducted only use a small subset of existing RL algorithms, limiting the paper's ability to draw broader conclusions about RL's applicability to sustainable energy tasks.
- The discussion on experiments, multi-agent settings, and physical constraints is relegated to the appendix, which may limit the paper's impact.
- Figures need some work: fonts are too small compared to the main text, some acronyms are not explicitly described anywhere in the paper, some are too small to analyze (e.g., Figures 1 and 2). Figure 5 in particular needs work as there is a lot of heterogeneity among the figures and the use of violin plots complicates the data visualization.

**Relation To Prior Work:**

There is a clear discussion of how this manuscript relates to and builds upon prior work.

**Summary And Contributions:**

The paper presents SustainGym, a suite of five RL environments designed to provide standardized benchmarks for the testing of RL algorithms on sustainable energy system tasks. The authors developed the environments to simulate real-world sustainability scenarios, including electric vehicle charging and carbon-aware data center job scheduling. They emphasized the representation of distribution shifts and multi-agent settings, which often appear in real-world situations but have been underrepresented in traditional RL benchmarks. The authors provide a comprehensive performance test using standard RL algorithms and identify areas where performance could be improved.

---

> ### Author Response · Authors · 2023-08-25
> **Part 1 of 2**
>
> Thank you for taking the time to provide thoughtful feedback. We have revised our manuscript based on your insights.
>
> > While the authors provide a useful introduction to the five environments, they could offer more justifications for their design decisions.
>
> We have updated the descriptions of each environment to emphasize what systems they are modeled after. All of our environments are either based on a particular real-world system (EVChargingEnv, DataCenterEnv, and CogenEnv) or are modeled after industry-standard test systems (ElectricityMarketEnv and BuildingEnv). Details of how representative each environment is, and the limitations of what they model, are now provided in the appendix.
>
> > The experiments conducted only use a small subset of existing RL algorithms, limiting the paper's ability to draw broader conclusions about RL's applicability to sustainable energy tasks.
>
> Indeed a thorough evaluation of RL algorithms for SustainGym is important, and we view that as very important future work. The main focus of our current work is the thoughtful design of high-quality RL gym environments for benchmarking energy systems. The availability of SustainGym will enable studying existing RL and MARL algorithms (and many other research questions) by the entire research community.
>
> Of course, we are also interested in running more experiments ourselves. We plan to do so once API support for the latest version of gymnasium API is supported by leading off-the-shelf RL libraries (StableBaselines3 and RLLib).
>
> We highlight that even with the subset of existing RL algorithms that we have tested, it is clear that off-the-shelf RL algorithms (such as PPO, SAC, and/or DQN) that perform well on the standard OpenAI gym environments may not perform well on sustainable energy tasks.
>
> > The discussion on experiments, multi-agent settings, and physical constraints is relegated to the appendix, which may limit the paper's impact.
> > [...] However, it could be improved by a more detailed discussion of multi-agent settings and physical constraints within the main body of the paper, rather than in the appendix.
>
> We appreciate your feedback and have moved more discussion of the experiments, multi-agent settings, and physical constraints into the main paper. Due to the page limit, this has meant that we had to move some of the existing environment descriptions into the appendix.
>
> > Figures need some work …
>
> Thank you for your feedback. We have updated some of the figures to be more visually consistent and will continue to improve the readability of the paper.
>
> > The discussion is very brief and I would expect to see some acknowledgment of the limitations of SustainGym. An obvious limitation is the lack of consideration of other sustainability dimensions. Sustainability goes beyond considerations of just energy and CO2 emissions and should consider factors such as water usage, eutrophication potential, etc., where possible. [...] Discussion of the limitations of SustainGym and the representativeness of its environments is also required.
>
> We absolutely agree that other sustainability dimensions beyond energy and CO2 are also extremely important to model. The reason we focus on CO2 and energy consumption is because we have in-house access to energy system and decarbonization domain experts that we consulted with in the process of building our environments. We use accurate sources of marginal CO2 emissions data as well as estimated prices for carbon (via carbon markets) which align economic incentives with sustainability objectives. Thus, we are proud that our energy and CO2 reward functions are representative and realistic.
>
> For our paper and website, we can add a more comprehensive discussion of broader sustainability objectives, and can highlight adjacent work such as ConservationGym. We have added a new “limitations” paragraph in the conclusion to highlight this point.

---

> ### Author Response · Authors · 2023-08-25
> **Part 2 of 2**
>
> > Another limitation is the representativeness of these sustainable energy tasks. [...] Discussion of the limitations of SustainGym and the representativeness of its environments is also required.
>
> Thank you for this feedback. As mentioned in our global response to all reviewers, the 5 environments currently available in SustainGym are meant to be representative of different energy-related control tasks. Some are based on a specific real-world system, whereas others represent a generic system. We have revised our submission to highlight these characteristics.
>
> As an example, EVChargingEnv is based on the actual EV charging networks in place at Caltech and the Jet Propulsion Laboratory (JPL), both located in Pasadena, California, U.S.A. EVChargingEnv uses a “digital twin” of these networks, called ACN-Sim, as well as real historical EV charging data, in the simulation of the RL environment. While different EV charging networks may have different constraints, the adaptive EV charging problem is similar across all networks. ACN-Sim (and therefore EVChargingEnv) can also be extended to model other charging networks as well.
>
> We have revised our manuscript to specifically highlight the generalizability and limitations of each of our environments in the appendix.

---

> > ### Comment · Reviewer_E4iv · 2023-08-30
> >
> > Thank you for your detailed responses. I have increased my score based on the changes made, particularly the introduction of a section on limitations. However, my concerns about figures have not been adequately addressed.

---

> ### Author Response · Authors · 2023-08-30
> **Updated Figures**
>
> We thank the reviewer for feedback and suggestions for our figures. We have updated the figures in response to improve clarity and readability.
>
> > Figures need some work: fonts are too small compared to the main text [...] some are too small to analyze (e.g., Figures 1 and 2).
>
> We have made updates to all of our figures (especially Figures 1, 2, and 3) to improve readability. Figures and fonts have been enlarged.
>
> > some acronyms are not explicitly described anywhere in the paper
>
> All acronyms are now explicitly described in the paper. Due to space constraints, certain acronyms such as A2C and DDPG are only described in the appendix instead of the main body of the paper. However, A2C and DDPG are such widely-used and cited RL algorithms that we do not believe that this will be an issue for most readers of our paper.
>
> > Figure 5 in particular needs work as there is a lot of heterogeneity among the figures and the use of violin plots complicates the data visualization.
>
> We believe that the reviewer intended to write Figure 4, instead of Figure 5. We have improved Figure 4 (specifically Figure 4e) to improve homogeneity and will continue making improvements as we run more experiments (see note about gymnasium/StableBaslines3/RLLib API issues in our earlier comment, "Part 1 of 2"). We use violin plots in order to convey that an algorithm's return on an environment is a random quantity and is therefore best represented by a distribution. The randomness comes from both randomness in the algorithm's policy and randomness in the environments. That said, if the reviewer has other suggestions for how to best communicate this point, we welcome discussion and advice.

---

### Official Review · Reviewer_pDRB · 2023-07-20

**Rating:** 6
**Confidence:** 3
**Clarity:** The paper is clearly written.

**Strengths:**

* Sustainability is a paramount topic, and exploring how to harness RL to tackle sustainability tasks is a truly inspiring direction.
* The introduced environments are built upon realistic sustainability tasks, offering significant practical value.
* The descriptions of the environments are clear and detailed.
* Overall, the paper is clearly written and easy to follow.

**Additional Feedback:**

No

**Correctness:**

The environments are designed appropriately. Expanding the scope of experiments beyond the mere evaluation of off-the-shelf RL algorithms would further enhance the benchmark's utility and showcase its potential for promoting future research in addressing real-world challenges.

**Documentation:**

A detailed README file is provided to give instructions on how to use the benchmark. There are detailed descriptions for each environment in the paper, though the documentation page only covers 2 environments.

**Ethics:**

Since the benchmark is about real-world sustainability tasks, it would be better to add discussions on societal impacts.

**Limitations:**

The limitations are discussed in Section 4.

**Opportunities For Improvement:**

* While I appreciate the detailed descriptions in Section 2, I think it would be better to have a more compact version in the main text and leave the full descriptions to the appendix or the documentation. The experiments can be expanded. For example, more analysis of how RL agents perform in each task would be helpful.
* I have some questions about Figure 4:
  * Why does the "random" baseline only appear in some subplots?
  * If I understand it correctly, the authors use blue color for the results under no distribution shift (except subplot-b). But this is only explicitly shown in the legend of subplot-d. In other subplots, the legend shows the concrete settings. I would suggest using "shift / no shift" for all subplots (and maybe using the same color for subplot-b). Readers can understand the figure without going back to Table 2.


**Relation To Prior Work:**

Related works have been clearly discussed.

**Summary And Contributions:**

This paper introduces SustainGym, a suite of 5 environments modeling realistic sustainable energy system tasks. The environments feature realistic distribution shifts and multi-agent settings. The authors evaluate several standard off-the-shelf RL algorithms on SustainGym. They highlight the challenges and significant room for improvement for applying RL in real-life sustainability tasks.

---

> ### Author Response · Authors · 2023-08-25
> **Author Response**
>
> Thank you for taking the time to provide thoughtful feedback. We have revised our manuscript based on your insights.
>
> > While I appreciate the detailed descriptions in Section 2, I think it would be better to have a more compact version in the main text and leave the full descriptions to the appendix or the documentation. The experiments can be expanded. For example, more analysis of how RL agents perform in each task would be helpful.
>
> We appreciate your feedback and have moved more discussion of the experiments, multi-agent settings, and physical constraints into the main paper. We have also moved some details of the environment descriptions into the appendix.
>
> > I have some questions about Figure 4: Why does the "random" baseline only appear in some subplots? If I understand it correctly, the authors use blue color for the results under no distribution shift (except subplot-b). But this is only explicitly shown in the legend of subplot-d. In other subplots, the legend shows the concrete settings. I would suggest using "shift / no shift" for all subplots (and maybe using the same color for subplot-b). Readers can understand the figure without going back to Table 2.
>
> Figure 4 compares the performance on the shifted environment between RL algorithms trained on the original environment (blue) and RL algorithms trained on the shifted environment (orange). We have revised the figure caption to be clearer, and the colors for Figure 4b will be updated. We will also update Figure 4 to include a random baseline in all subplots.

---

### Official Review · Reviewer_P6Jo · 2023-07-20
**Review of SustainGym**

**Rating:** 7
**Confidence:** 4

**Strengths:**

The paper clearly describes each of these five environments.The definition of each enviroment is clearly stated and justified, in particular the fact of taking distribution shifts into account.

**Additional Feedback:**

No particular feedback.

**Clarity:**

The paper is well organized and clearly written. Its comprehension are thus made easy.

**Correctness:**

The paper explains in a correct way the various environments of the SustainGym suite.

**Documentation:**

The data used are clearly documented, as well as the algorithms and the related work.

**Limitations:**

Some limitations are given in Section 4. However, it would be interesting to split the limitations in two parts: those related to the environments, and those due to the algorithms.

**Opportunities For Improvement:**

First, explaining in more details how those five environments really make a suite, beyond the fact that they are dedicated to sustainability matters, would give more impact to SustainGym.
Second, on line 39, the "two categories" are not clearly documented. The reader has to believe you.
Third, the choice of the off-the-shelf RL algorithms is clearly made for illustrative purposes. However, a reader might wonder to which extend the five environments are performant for any type of MARL algorithm.
Appendices are not provided.Therefore, the results of the experiments are quite difficult to interpret. Which does not mean that the environments are not useful.

**Relation To Prior Work:**

The positioning of SustainGym with respect to previous related work is quite clear.

**Summary And Contributions:**

The paper presents SustainGym, which is a suite of five environments that are dedicated to benchmarking MARL algorithms for these specific issues.

---

> ### Author Response · Authors · 2023-08-25
> **Author Response**
>
> Thank you for taking the time to provide thoughtful feedback. We have revised our manuscript based on your insights.
>
> > First, explaining in more details how those five environments really make a suite, beyond the fact that they are dedicated to sustainability matters, would give more impact to SustainGym.
>
> The 5 environments in SustainGym comprise a “suite” in the sense that they feature certain similar characteristics while also testing RL algorithms across a wide range of challenges.The main similarities in terms of design are:
> real-world or representative environments for energy systems
> reward functions based on reducing CO2 and/or fuel consumption
> environments feature realistic distribution shifts
> easy to use with both the popular RLLib and StableBaselines3 libraries
> The diverse set of challenges for RL algorithms covered by the gyms are:
> varying levels and types of distribution shifts
> single- and multi-agent RL. While the current MARL environments in SustainGym are all cooperative, we plan on extending the ElectricityMarketEnv to the competitive MARL setting as well.
> actions with constraints
> Notably, our experimental results should already be interesting to RL researchers.
>
> > Second, on line 39, the "two categories" are not clearly documented. The reader has to believe you.
>
> We have improved the readability of this part of the paper.  Specifically, we have added a paragraph comparing the distribution shift setting in SustainGym with nonstationary RL environments and distributionally robust RL.
>
> > Third, the choice of the off-the-shelf RL algorithms is clearly made for illustrative purposes. However, a reader might wonder to which extend the five environments are performant for any type of MARL algorithm. Appendices are not provided. Therefore, the results of the experiments are quite difficult to interpret. Which does not mean that the environments are not useful.
>
> Indeed a thorough evaluation of MARL for SustainGym is important, and we view that as very important future work. The main focus of our current work is the thoughtful design of high-quality RL gym environments for benchmarking energy systems. The availability of SustainGym will enable studying MARL (and many other research questions) by the entire research community.  Also, the specific experiments and models are detailed in the appendix sections corresponding to each environment.
>
> Of course, we are also interested in running more experiments ourselves. We plan to do so once API support for the latest version of gymnasium API is supported by leading off-the-shelf RL libraries (StableBaselines3 and RLLib).

---

> > ### Comment · Reviewer_P6Jo · 2023-08-30
> >
> > Thank you for your responses and improvements.

---

### Official Review · Reviewer_4BfE · 2023-07-21
**A broad set of simulated environments for training RL agents, that needs some discussion on limitations and generalizability**

**Rating:** 6
**Confidence:** 4
**Clarity:** The paper is well written and clear w…

**Strengths:**

- The application of sustainable energy improvement is a signifiant and valuable one that requires further optimization. This helps in that endeavor.
- The Gym is released on Github
- The five environments cover a range of settings within sustainable infrastructure

**Additional Feedback:**

- The figure captions are tiny and require a lot of zoom to read. A good rule of thumb is that the minimum figure size is not significantly smaller than the normal paragraph text in the document to ensure accessibility.
- Make it clear what the application is. For example, in the EV charging, is it the charging station that would want to optimize using this environment or the EV owner? Or both? Others?
- The equation on line 145 could use some spacing improvements
- Figure 4e uses a different background which looks out of place visually

**Correctness:**

The environments are implemented as stated in the contribution. However, as with all simulations of the real world, there are assumption and limitations which must be acknowledged in order to correctly interpret the outputs. The authors should add more discussion on limitations and assumptions (see above too).

**Documentation:**

I would have preferred to see more documentation and how-to guides on the Github repo accompanying the paper. It will significantly ease adoption and improve the usability of this work.

**Ethics:**

I encourage the authors to consider the ethical effects of this work. There are many questions that need to be considered, including how does dynamic pricing/electricity charging affect low-income individuals?

**Limitations:**

I disagree that the authors have discussed the limitations of their work, the checklist states this is in Section 4. There are a number of limitations worth discussing, including but not limited to:
- Some distribution shifts are modelled, others are not. There are many sources of distribution shift in the real world, which ones do you not capture? How do you estimate the effect of ignoring those?
- The environments are based on specific implementations. E.g. the EV network uses data from Caltech and JPL charging stations. How does this generalize to other EV station? A discussion on how representative this is would be very welcomed.
- Likewise, the generalizability of the other environments. Different electricity markets have very different rules for managing loads and offer spreads. How general is this implementation? Is it modelled on the California market? Are there any major world markets that are incompatible with this implementation?

**Opportunities For Improvement:**

There are three major factors where I see possible improvement:
- The level of application specificity is not covered. Is this based specifically on the real-world cases with additional distribution shift on top or is it generalizable to most settings?
- Documentation covering the code can be improve for accessibility. Is it not very easy to get started. A simple guide on how to get started using each environment would be a valuable addition.
- The limitations of the work are not discussed sufficiently.

**Relation To Prior Work:**

Line 70 says an extended related works section is found in Section 4. This does not seem to be the case. There is no extended related work section.

The current discussion of related work is just about sufficient. More would be helpful but not required in my opinion, given the topic of work.

**Summary And Contributions:**

The authors present a set of environments to train and test reinforcement learning agents in the field of sustainable energy systems. The set, SustainGym, includes tasks of: EV charging, electricity market, datacenter carbon optimization, cogeneration plants and thermal building control. The sim also incorporates changes (distribution shifts) in demand and changes in environmental factors such as the make up of powerplants in the setting.

They then test a number of existing RL approaches using the sim and find that their performance is lacking.

---

> ### Author Response · Authors · 2023-08-25
> **Part 1 of 2**
>
> Thank you for taking the time to provide thoughtful feedback. We have revised our manuscript based on your insights.
>
> > The level of application specificity is not covered. Is this based specifically on the real-world cases with additional distribution shift on top or is it generalizable to most settings?
> > [...]
> > The limitations of the work are not discussed sufficiently. Some distribution shifts are modelled, others are not. There are many sources of distribution shift in the real world, which ones do you not capture? How do you estimate the effect of ignoring those?
>
> We have updated the manuscript to detail which environments are based on specific real-world systems (EVChargingEnv, DataCenterEnv, CogenEnv) and which environments are based on industry standard test cases (ElectricityMarketEnv and BuildingEnv). We have also revised our paper to include more details about the generalizability and specificity of our environments.
>
> Regarding distribution shift, we recognize that there are many unmodeled sources of distribution shift, and we have updated our manuscript to reflect that. Our specific choices of distribution shift were chosen because they were either based on real-world distribution shifts seen in our data (EVChargingEnv, ElectricityMarketEnv, DataCenterEnv) or based on suggestions from industry experts that we spoke with (CogenEnv, BuildingEnv).
>
> > Documentation covering the code can be improve for accessibility. Is it not very easy to get started. A simple guide on how to get started using each environment would be a valuable addition.
>
> We have updated our website with getting started guides for most of the environments and are continuously improving our documentation to make it easier to get started using SustainGym. Among our goals is to eventually release a PyPi package for SustainGym. However, we are waiting for the gymnasium API (which is replacing the older OpenAI gym API) to stabilize and the standard RL libraries (StableBaselines3 and RLLib) to support the latest gymnasium before we release our PyPi package.
>
> > The environments are based on specific implementations. E.g. the EV network uses data from Caltech and JPL charging stations. How does this generalize to other EV station? A discussion on how representative this is would be very welcomed. Likewise, the generalizability of the other environments. Different electricity markets have very different rules for managing loads and offer spreads. How general is this implementation? Is it modelled on the California market? Are there any major world markets that are incompatible with this implementation?
>
> The 5 environments currently available in SustainGym are meant to be representative of different energy-related control tasks. Some are based on a specific real-world system, whereas others represent a generic system. We have revised our submission to highlight these characteristics.
>
> Specifically, EVChargingEnv is based on the actual EV charging networks in place at Caltech and the Jet Propulsion Laboratory (JPL), both located in Pasadena, California, U.S.A. EVChargingEnv uses a “digital twin” of these networks, called ACN-Sim, as well as real historical EV charging data, in the simulation of the RL environment. While different EV charging networks may have different constraints, the adaptive EV charging problem is similar across all networks. ACN-Sim (and therefore EVChargingEnv) can also be extended to model other charging networks as well.
>
> ElectricityMarketEnv is based on the IEEE RTS-GMLC test case, proposed in 2019. The RTS-GMLC test case was designed to be representative of a modern transmission network located in the southwestern U.S., featuring a variety of renewable and distributed generators as well as representative electricity load profiles. ElectricityMarketEnv can also be configured to more closely match a particular network. However, as you correctly note, different electricity markets use very different rules for managing loads and offer spreads. These differences are especially stark when comparing U.S. electricity markets with those in Asia, for example. Our team has expertise with U.S. energy markets and therefore we do not make any claim about whether our RL environment is compatible with electricity markets in other countries. We welcome collaborations with experts in other countries to help us increase the flexibility of our environment to electricity markets outside of the U.S.

---

> ### Author Response · Authors · 2023-08-25
> **Part 2 of 2**
>
> > There are many questions that need to be considered, including how does dynamic pricing/electricity charging affect low-income individuals?
>
> Thank you for highlighting this potential consideration. Because Caltech and JPL have fixed-rate electricity prices and since EVChargingEnv is based on EV charging networks at Caltech and JPL, EVChargingEnv does not use dynamic pricing. We have updated the manuscript to emphasize this. Furthermore, because of power and network constraints, previous research (https://ieeexplore.ieee.org/document/9409126) has shown that non-adaptive (e.g., greedy) charging algorithms do a much poorer job at charging EVs. With adaptive RL and MPC algorithms, most EVs that come to Caltech and JPL are able to reach a full charge.
>
> As for other environments, we have not identified specific ethical considerations, as our simulations are based on either existing real-world or widely-used industry-standard test cases. However, we welcome your thoughts if you believe that we have overlooked other ethical considerations.

---

### Author Response · Authors · 2023-08-25
**General Response to Reviewers**

We thank all reviewers for your time and helpful comments. Here, we hope to address the general and shared concerns. We have also revised our submitted manuscript, with major changes shown in blue.

###  Generalizability and RL environment design decisions

The 5 environments currently available in SustainGym are meant to be representative of different energy-related control tasks. Some are based on a specific real-world system, whereas others represent a generic system. We have revised our submission to highlight these characteristics and updated Table 1 to include this information. In the appendix, we have added separate sections on “Representativeness and Generalizability” and “Limitations” for all 5 of the SustainGym environments.

As an example, EVChargingEnv is based on the actual EV charging networks in place at Caltech and the Jet Propulsion Laboratory (JPL), both located in Pasadena, California, U.S.A. EVChargingEnv uses a “digital twin” of these networks, called ACN-Sim, as well as real historical EV charging data. While different EV charging networks may have different constraints, the adaptive EV charging problem is similar across all networks. ACN-Sim (and therefore EVChargingEnv) can also be extended to model other charging networks as well.

Overall, we feel the design of our environments strikes a good balance between generalizability and leveraging real-world data where possible.

### Formatting

We appreciate your feedback and have moved more discussion of the experiments, multi-agent settings, and physical constraints into the main paper. Due to the page limit, this has meant that we had to move some of the existing environment descriptions into the appendix. We have also updated some of the figures to be more visually consistent and will continue to improve the readability of the paper.

### Documentation

We have updated the SustainGym website (https://chrisyeh96.github.io/sustaingym/) with better documentation for all 5 environments. We plan on continuing to improve the documentation as we continue to add more functionality and customizability to our environments.

---

### Decision · Program_Chairs · 2023-09-22

**Decision:**

Accept (Poster)

**Comment:**

### Summary And Contributions:
The authors present a set of environments to train and test reinforcement learning agents in the field of sustainable energy systems. The set, SustainGym, includes tasks of: EV charging, electricity market, datacenter carbon optimization, cogeneration plants and thermal building control. The sim also incorporates changes (distribution shifts) in demand and changes in environmental factors.

### Strengths:
- The application of sustainable energy improvement is a significant and valuable one that requires further optimization. This helps in that endeavor.
- The definition of each environment is clearly stated and justified, in particular the fact of taking distribution shifts into account.
- The Gym is released on Github.

### Opportunities For Improvement:
- The experiments conducted only use a small subset of existing RL algorithms, limiting the paper's ability to draw broader conclusions about RL's applicability to sustainable energy tasks.
- A more detailed discussion of multi-agent settings and physical constraints.
- Future work: how to expand the current 5 environments to incorporate and model other factors.

### Clarity:
- The paper is well written and clear with good sections and flow.

Overall, the reviewers agreed that SustainGym is a good idea, well formulated, clearly defining and new set of environments focused on sustainable energy systems.

Since the authors have incorporated the feedback provided by the reviewers and improved the paper sustainably as reviewers asked for, I'm recommending acceptance.